# Hierarchical Decision Making with Structured Policies: A Principled Design via Inverse Optimization

## Abstract

Hierarchical decision-making frameworks are pivotal for addressing complex control tasks, enabling agents to decompose intricate problems into manageable subgoals. Despite their promise, existing hierarchical policies face critical limitations: (i) reinforcement learning (RL)-based methods struggle to guarantee strict constraint satisfaction, and (ii) optimization-based approaches often rely on myopic and computationally prohibitive formulations. To reconcile these trade-offs, hierarchical RL-OC architectures have emerged as a promising paradigm. However, the formulation of the lower-level optimization within these frameworks remains underexplored, often relying on heuristic or myopic objectives. In this work, we propose a principled framework that systematically integrates upper-level goal abstraction with structured lower-level decision making. We adopt an inverse optimization approach to inform the structure of the lower-level problem from expert demonstrations, ensuring that the objective of the lower-level policy remains aligned with the overall long-term task goal. To validate the approach, our framework is evaluated on distinct decision making tasks: network-based resource allocation and continuous collision avoidance. Empirical results demonstrate that our method outperforms Learning-based OC, End-to-end RL and Representative Hierarchical RL in both efficiency and decision quality.

## 1 Introduction

Real-time decision-making in cyber-physical systems, such as robotics, autonomous driving, power grids, and transportation (Jendoubi & Bouffard, 2023; Zhou et al., 2024; Liang et al., 2025), is inherently challenging due to high-dimensional state and action spaces, nonlinear dynamics, and complex physical constraints. Existing solutions largely stem from optimal control (OC) and deep reinforcement learning (RL). OC-based methods aim to optimize system performance over infinite or long horizons while ensuring stability and feasibility. These methods are well-suited for safety-critical systems due to their theoretical guarantees, but may scale poorly in high-dimensional or nonlinear settings and often require accurate models. In contrast, RL-based approaches directly learns a policy from interactions with the environment, which scale well to complex tasks and do not require explicit modeling of complex system dynamics. Nevertheless, these approaches require extensive training and lack safety or constraint satisfaction guarantees due to their black-box nature. These trade-offs have motivated growing interest in combining OC-based and RL-based methods to exploit the strengths of both paradigms.

A promising approach for combining OC- and RL-based methods is through a hierarchical architecture that decomposes decision-making into two sequential subproblems (Lew et al., 2023; Karnchanachari et al., 2020). The upper-level employs an RL policy for strategic planning, such as generating subgoals, while the lower level uses OC to ensure safe and feasible execution. This hierarchical architecture not only enhances scalability and feasibility but also aligns with human cognition, as humans tend to perform abstract planning guided by intrinsic motivation, grounded by fast, lower-level execution.

Despite the promise of hierarchical RL–OC frameworks, the formulation of the lower-level optimization problem remains underexplored. The lower-level controller must be both computationally efficient and aligned with the upper-level goals, since a poorly designed formulation may inad-

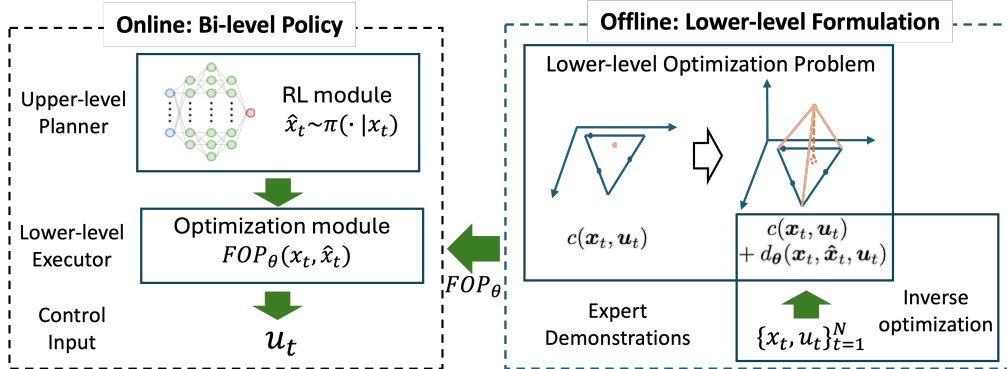

Figure 1: We propose an RL-OC hierarchical decision-making framework with lower-level policy informed by inverse optimization.

vertently exclude high-quality solutions. Existing approaches have several limitations. First, most hierarchical methods adopt long-horizon OC formulations at the lower level to preserve stability and feasibility guarantees (Song & Scaramuzza; Cheng et al., 2024; Landgraf et al., 2022). However, such long-horizon MPC formulations are well known to incur prohibitive computational complexity for real-time applications (Karamanakos et al., 2014; Krishnamoorthy et al., 2020). Second, recent efforts leverage single-step OC to enhance computation efficiency by directly generating the value or constraints on the desired next state (Gammelli et al., 2023; Schmidt et al., 2024). Nevertheless, these formulations typically rely on myopic objectives without an appropriately designed formulation. As established in classical optimal control theory, short horizons without accurate terminal value approximations cause solutions to overlook longer-term impacts, potentially leading to suboptimal trajectories (Rawlings et al., 2020; Lowrey et al., 2018). The above challenges highlight the importance of formulating the lower-level optimization problem in a way that reduces sub-optimality while simultaneously ensuring tractable computational complexity.

To address these limitations, we propose an inverse optimization framework to guide the design of the lower-level policy from a handful of demonstrations, which provide valuable insights (Figure 1). We cast the design of the lower-level cost function as an inverse optimization problem. For a special class of lower-optimization problems with linear cost functions, we provide a theoretical characterization of the conditions under which the expert demonstrations are optimal. Once the formulation is established, we employ efficient methods to solve the inverse problem, determining the exact mathematical formulation that best fits the observed data. To validate the effectiveness of our approach, we demonstrate our method on the problem of autonomous vehicle rebalancing, supply chain inventory management and mobile robot navigation. The improvements in learning the formulation are validated from multiple perspectives. In light of the above discussion, we summarize the main contributions of this work as follows:

- We propose an inverse optimization-based approach to systematically design the lower-level control policy within a hierarchical RL-OC framework.

- We provide theoretical analysis for a special class of problems with broad applications, proposing a tractable cost structure and efficient inverse optimization formulation which ensures inverse-feasibility, forward-stability, and computational tractability.

- We demonstrate the effectiveness of the proposed framework on several scenarios from different fields, showcasing its practical relevance and potential impact in real-world applications.

## 2 RELATED WORK

This work is related to the literature on hierarchical structured control policies. Depending on whether learning-based or model-based approaches are used at each level, existing works can be broadly classified into two categories: (i) hierarchical reinforcement learning (HRL) that employs RL at each level, and (ii) learning-based optimal control, e.g., frameworks integrating RL and MPC,

whereby a upper-level policy learns desired states or goals, and a simplified lower-level MPC ensures safe and feasible execution.

**Hierarchical Reinforcement Learning** HRL decomposes a complex, difficult-to-solve problem into multiple simpler, smaller problems by setting subgoals (Kulkarni et al., 2016; Vezhnevets et al., 2017; Ma et al., 2021; Xie et al., 2021; Eppe et al., 2022; Qi et al., 2022; Huang et al., 2022; Gu et al., 2023; Mao et al., 2024; Luo et al., 2024; Zhang et al., 2024b; Hirt et al., 2024). We focus on how hierarchical policies utilize various forms of intrinsic motivation by setting subgoals. Naveed et al. (2021) develop a hierarchical reinforcement learning framework for autonomous vehicle trajectory planning, where a upper-level policy selects maneuver options, and a lower-level planner generates waypoints accordingly. Vezhnevets et al. (2017) applies Feudal Networks for hierarchical reinforcement learning. The manager module sets abstract goals that are conveyed to and enacted by the Worker module. Another common way to set goals is to consider the desired states. In Nachum et al. (2018), the author sets the upper-level actions to be goal states and reward the lower-level policy for performing actions that yield an observation close to matching the desired goal. In recent years, a growing body of work has investigated how to define subgoals and how to search efficiently within the subgoal space (Liu et al., 2021; Ma et al., 2023). Nevertheless, these studies have paid limited attention to the relationship between subgoals and the agent's final actions. Moreover, as previously discussed, incorporating constraints to ensure safety strictly is challenging in reinforcement learning framework. In next section, we focus on how learning, especially reinforcement learning, interplays with optimization in the previous literature.

**Learning-based OC.** In the control community, various real-world control problems are solved by using learning-based OC. In many existing works, learning-based methods are often applied to learn cost functions or system dynamics (Lenz et al., 2015; Coulson et al., 2019; Hewing et al., 2020; Dogan et al., 2023; Zhang et al., 2024a; Lu et al., 2024; Zhang et al., 2024a; Dinkla et al., 2026). However, solving optimal control problems in real time still poses challenges when long control horizons are used, due to the high dimensionality of variables and the complexity of constraints. Prior work has attempted to reduce the computational burden by approximating the long-horizon MPC problem with a single-step formulation and learning terminal cost to alleviate myopic behavior (Abdufattokhov et al., 2021; Alsmeier et al., 2024). However, the learning paradigm of these methods typically optimizes a surrogate objective (e.g., immediate fit to reference) rather than the final cumulative rewards directly (Zhang et al., 2024a). Instead, Gammelli et al. (2023) propose to leverage reinforcement learning to learn upper-level actions to shorten the control horizon of network flow control problem. In this framework, reinforcement learning is applied to provide reference trajectories that guide the lower-level executor toward maximizing cumulative rewards. However, the lower-level optimization problem still hinges on myopic objectives and does not rely on an explicitly and appropriately designed formulation. In addition, Schmidt et al. (2024) study hierarchical RL-OC in an offline setting and focus on generating upper-level subgoals. Their framework assumes that the offline data are optimal for the lower-level control problem, which can be sometimes unrealistic and leaves the design of the lower-level OC formulation unaddressed.

Related to this line of research, our work investigates how to formulate the lower-level optimization problem in hierarchical RL-OC framework to alleviate sub-optimality issues stemming from structural myopia.

## 3 METHODOLOGY

### 3.1 PROBLEM SETTING AND PRELIMINARY

Let us consider a general multi-step decision-making problem formulated in (1).

$$
\begin{aligned}
\min_{\{\boldsymbol{u}_t\}_{t=0}^{\infty}} \ &\limsup_{T \to \infty} \frac{1}{T} \sum_{t=0}^{T-1} c(\boldsymbol{x}_t, \boldsymbol{u}_t) \\
\text{s.t.} \quad &x_{t+1} = f(\boldsymbol{x}_t) + g(\boldsymbol{x}_t)\boldsymbol{u}_t, \quad \forall t \geq 0 \\
&\boldsymbol{x}_t \in \mathcal{X}_t, \quad \forall t \geq 0 \\
&\boldsymbol{u}_t \in \mathcal{U}_t, \quad \forall t \geq 0
\end{aligned}
\tag{1}
$$

where $\boldsymbol{x}_t \in \mathbb{R}^n$ is the system state at time step $t$, $\boldsymbol{u}_t \in \mathbb{R}^m$ is the control input, and $c(\boldsymbol{x}_t, \boldsymbol{u}_t)$ is the stage cost function assumed to be convex. The system dynamics are assumed to be control-affine (i.e., linear in $\boldsymbol{u}_t$) with functions $f(\cdot)$ and $g(\cdot)$, which cover a wide range of applications in cyber-physical systems. The feasible sets $\mathcal{X}_t$ and $\mathcal{U}_t$ encode admissible states and inputs, respectively, which are assumed to be convex. The initial state $x_0$ is known.

Due to the infinite horizon and the presence of complex constraints, solving Problem (1) is often computationally intractable. A standard workaround is a finite-horizon approximation. However, OC problems with long time horizons can still be computationally challenging for large-scale systems, which do not satisfy the real-time requirements of practical applications. To address this issue, we generalize the bi-level decision-making framework proposed by Gammelli et al. (2023) to a more general problem setting.

## 3.2 Bi-level Framework

The hierarchical RL-OC framework is implemented as an end-to-end system shown in Problem (2).

$$\pi^* \in \arg\max_{\pi \in \Pi} \mathbb{E}_\tau \left[ \sum_{t=0}^{\infty} \gamma^t c(\boldsymbol{u}_t, \boldsymbol{x}_t) \right]$$
$$\text{s.t.} \quad h_t \sim \pi(h_t \mid \boldsymbol{x}_t)$$
$$\boldsymbol{u}_t = \text{FOP}(h_t, \boldsymbol{x}_t)$$

(2)

The overall policy $\pi^*$ composes an upper-level RL policy $\pi$ and the solution to a lower-level optimization problem FOP. The upper-level policy encodes task-relevant abstract information or goals to produce the intrinsic subgoal $h_t$. The lower-level optimization module receives both the intrinsic subgoal $h_t$ and the current state $\boldsymbol{x}_t$ to compute the control input $\boldsymbol{u}_t$ that is feasible.

*Remark* 1 (Practical requirements for RL-OC frameworks). There are two requirements for Problem (2). First, for real-time deployment, FOP should capture operational constraints and be computationally efficient to solve. Second, to enable stable training, the intrinsic subgoal $h_t$ is preferably low-dimensional.

Following Remark 1, a commonly used form of subgoals is a linear transformation of the desired next state $\boldsymbol{x}_t^{\text{des}}$ using a known matrix $C$, denoted by $\hat{\boldsymbol{x}}_t = C\boldsymbol{x}_t^{\text{des}}$ (Gammelli et al., 2023; Schmidt et al., 2024). This subgoal guides the system toward the desired state by inducing appropriate lower-level actions. Here, in scenarios with high state dimensions, the transformation matrix $C$ serves to compress the action space of the upper-level RL by mapping the high-dimensional system state to a lower-dimensional planning space. The transformation matrix $C$ is typically set as an identity matrix in scenarios with low state dimensions. Overall, the bi-level decision-making framework is given in (3) and (4).

$$\pi^* \in \arg\max_{\pi \in \Pi} \mathbb{E}_\tau \left[ \sum_{t=0}^{\infty} \gamma^t c(\boldsymbol{u}_t, \boldsymbol{x}_t) \right]$$
$$s.t. \quad \hat{\boldsymbol{x}}_t \sim \pi(\hat{\boldsymbol{x}}_t \mid \boldsymbol{x}_t)$$
$$\boldsymbol{u}_t \in \text{FOP}(\boldsymbol{x}_t, \hat{\boldsymbol{x}}_t)$$

(3)

$$\text{FOP}(\boldsymbol{x}_t, \hat{\boldsymbol{x}}_t) := \quad \arg\min_{\boldsymbol{u}_t} \ c(\boldsymbol{x}_t, \boldsymbol{u}_t)$$
$$\text{s.t.} \quad Cf(\boldsymbol{x}_t) + Cg(\boldsymbol{x}_t)\boldsymbol{u}_t = \hat{\boldsymbol{x}}_t,$$
$$f(\boldsymbol{x}_t) + g(\boldsymbol{x}_t)\boldsymbol{u}_t \in \mathcal{X}_{t+1}, \ \boldsymbol{u}_t \in \mathcal{U}_t$$

(4)

By leveraging intrinsic subgoals learned by the upper-level policy, the lower-level executor is able to make near-instantaneous, subgoal-conditioned decisions, which is particularly advantageous in time-sensitive or high-dimensional environments. However, the formulation of FOP still remains ambiguous in the bi-level framework, as discussed in Remark 2.

*Remark* 2 (Importance of proper lower-level formulations). We highlight two considerations for Problem (4). First, the transformed desired next state $\hat{\boldsymbol{x}}_t$ produced by the upper-level RL may be infeasible in practice, e.g., planned trajectories in robotics tasks, which require the design of a cost function to penalize violations. Such a cost function must be aligned with the overarching objective

of the decision-making problem to avoid sub-optimality. Second, even when $\hat{x}$ is feasible, the formulation of the lower-level optimization still requires careful design to mitigate myopic decisions, particularly when the transformation matrix $C$ reduces the state dimension. Specifically, there may exist multiple actions $\boldsymbol{u}_t$ that satisfy the constraint $Cf(\boldsymbol{x}_t) + Cg(\boldsymbol{x}_t)\boldsymbol{u}_t = \hat{\boldsymbol{x}}_t$ but yield different next states under the true dynamics, which will influence future rewards. Therefore, it is important to properly design the lower-level problem to align the action selection with the overarching objective.

In the next subsection, we will present an inverse optimization approach to inform the design of the lower-level optimization problem.

### 3.3 LEARNING PROBLEM FORMULATION: INVERSE OPTIMIZATION

#### 3.3.1 GENERAL FRAMEWORK

We assume access to a set of expert data $\{\boldsymbol{x}_t, \boldsymbol{u}_t\}_{t \in \mathcal{T}_e}$ generated by an existing decision-making policy, such as Model Predictive Control (MPC) or other high-quality heuristics ensuring strict safety guarantees, where $\mathcal{T}_e$ represents the set of time steps at which the dataset is collected. We make three remarks regarding the expert dataset. First, as high-quality solution methods may not be suitable for making real-time decisions in practice, the expert data can be derived offline rather than from real-world operations. Second, for the same reason, the dataset is assumed to be small, which makes supervised learning methods such as imitation learning less suitable. Third, $\mathcal{T}_e$ is chosen to cover diverse operating conditions and does not have to consist of consecutive time steps.

The core of our approach is to recover latent structure encoded in the expert data. Specifically, we parameterize the lower-level optimization problem as

$$\text{FOP}_{\boldsymbol{\theta}}(\boldsymbol{x}_t, \hat{\boldsymbol{x}}_t) := \arg\min_{\boldsymbol{u}_t}\{c(\boldsymbol{x}_t, \boldsymbol{u}_t) + d_{\boldsymbol{\theta}}(\boldsymbol{x}_t, \hat{\boldsymbol{x}}_t, \boldsymbol{u}_t) \mid \boldsymbol{a}_t + B_t\boldsymbol{u}_t = \hat{\boldsymbol{x}}_t, \boldsymbol{u}_t \in \widehat{\mathcal{U}}_t\} \quad (5)$$

where $\boldsymbol{a}_t := Cf(\boldsymbol{x}_t)$, $B_t := Cg(\boldsymbol{x}_t)$, and $\widehat{\mathcal{U}}_t := \mathcal{U}_t \cap \{\boldsymbol{u}_t | f(\boldsymbol{x}_t) + g(\boldsymbol{x}_t)\boldsymbol{u}_t \in \mathcal{X}_{t+1}\}$. The term $d_{\boldsymbol{\theta}}(\cdot)$ parameterized by parameter $\boldsymbol{\theta}$ is assumed to be convex and will be properly designed to align the lower-level optimization problem to the objective of the overarching decision-making problem.

We then convert the design of the lower-level optimization problem into estimating $\boldsymbol{\theta}$ from the expert dataset. Specifically, we aim to find $\boldsymbol{\theta}$ via inverse optimization such that for each expert pair $(\boldsymbol{x}_t, \boldsymbol{u}_t)$, the action $\boldsymbol{u}_t$ is (approximately) optimal for $\text{FOP}_{\boldsymbol{\theta}}(\boldsymbol{x}_t, \hat{\boldsymbol{x}}_t)$ under some subgoal $\hat{\boldsymbol{x}}_t$. Specifically, let $\mathcal{U}^{\text{opt}}(\boldsymbol{\theta}, \boldsymbol{x}_t, \hat{\boldsymbol{x}}_t)$ denote the optimal solution set corresponding to parameter $\boldsymbol{\theta}$, state $\boldsymbol{x}_t$, and subgoal $\hat{\boldsymbol{x}}_t$. In practice, since the lower-level optimization is convex, $\mathcal{U}^{\text{opt}}(\boldsymbol{\theta}, \boldsymbol{x}_t, \hat{\boldsymbol{x}}_t)$ can be characterized by the Karush–Kuhn–Tucker (KKT) conditions. Then, the inverse optimization problem can be formulated as

$$\min_{\boldsymbol{\theta}}\left\{\kappa h(\boldsymbol{\theta}) + \frac{1}{|\mathcal{T}_e|}\sum_{t \in \mathcal{T}_e}\ell\left(\boldsymbol{u}_t, \mathcal{U}^{\text{opt}}(\boldsymbol{\theta}, \boldsymbol{x}_t, \hat{\boldsymbol{x}}_t)\right) \,\middle|\, \boldsymbol{\theta} \in \boldsymbol{\Theta}\right\}. \quad (6)$$

where the objective function is a weighted combination of two parts: (i) the sum of losses with each loss $\ell(\boldsymbol{u}_t, \mathcal{U}_t^{\text{opt}}(\boldsymbol{\theta}))$ indicating the deviation of expert action $\boldsymbol{u}_t$ from the optimal solution set $\mathcal{U}_t^{\text{opt}}(\boldsymbol{\theta})$, i.e., the sub-optimality of the expert action, and (ii) a user-defined, application-specific regularization term $h(\boldsymbol{\theta})$ representing prior information or user preference regarding $\boldsymbol{\theta}$.

*Remark* 3 (Inverse optimization for lower-level formulations). We make three remarks regarding the inverse optimization framework. First, we focus on learning the objective function of $\text{FOP}_{\boldsymbol{\theta}}$ rather than its constraints, as constraints are typically dictated by physical requirements. Second, without loss of generality, we assume the constraint $\boldsymbol{a}_t + B_t\boldsymbol{u}_t = \hat{\boldsymbol{x}}_t$ is feasible. If this is not the case, we can relax it as $\boldsymbol{a}_t + B_t\boldsymbol{u}_t = \hat{\boldsymbol{x}}_t + \boldsymbol{\epsilon}_t$, and augment the action, corresponding matrix, and cost function as $\tilde{\boldsymbol{u}}_t = [\boldsymbol{u}_t; \boldsymbol{\epsilon}_t]$, $\tilde{B}_t = [B_t, I]$, and $d_{\boldsymbol{\theta}}(\boldsymbol{x}_t, \hat{\boldsymbol{x}}_t, \tilde{\boldsymbol{u}}_t)$, respectively, which yields a lower-level optimizer of the same form. Third, we focus on common cases where the subgoal $\hat{\boldsymbol{x}}_t$ can be retrieved from expert data, such as the desired next state or state representations as mentioned above. For other cases where subgoals cannot be directly obtained, they can be introduced as additional latent variables $\{\hat{\boldsymbol{x}}_t\}_{t \in \mathcal{T}_e}$ and estimated jointly with $\boldsymbol{\theta}$.

Without loss of generality, we assume $\widetilde{\mathcal{U}}_t$ is a polytope represented by $\widetilde{\mathcal{U}}_t = \{u \mid H_t \boldsymbol{u} \leq \boldsymbol{b}\}$. Under this assumption, the learning problem for $\theta$ can be formulated as

$$\min_{\boldsymbol{\theta}, \{\boldsymbol{w}_t, \boldsymbol{\lambda}_t, \boldsymbol{\epsilon}_t\}_{t \in \mathcal{T}_e}} \quad \sum_{t \in \mathcal{T}_e} \|\boldsymbol{\epsilon}_t\|_2^2 + \kappa h(\boldsymbol{\theta}) \tag{7a}$$

$$\text{s.t.} \quad \boldsymbol{0} \in \partial_{\boldsymbol{u}} \left( c(\boldsymbol{x}_t, \boldsymbol{u}_t) + d_{\boldsymbol{\theta}}(\boldsymbol{x}_t, \hat{\boldsymbol{x}}_t, \boldsymbol{u}_t) \right) + \boldsymbol{\lambda}_t^T H_t + \boldsymbol{w}_t^T B_t, \ \forall t \tag{7b}$$

$$\boldsymbol{a}_t + B_t \boldsymbol{u}_t = \hat{\boldsymbol{x}}_t, \quad H_t \boldsymbol{u} \leq \boldsymbol{b}, \ \forall t \tag{7c}$$

$$\boldsymbol{\lambda}_t \geq 0, \ \forall t \tag{7d}$$

$$\varphi(\boldsymbol{u}_t, \boldsymbol{\lambda}_t, \boldsymbol{w}_t) + \boldsymbol{\epsilon}_t = c(\boldsymbol{x}_t, \boldsymbol{u}_t) + d_{\boldsymbol{\theta}}(\boldsymbol{x}_t, \hat{\boldsymbol{x}}_t, \boldsymbol{u}_t), \ \forall t \tag{7e}$$

where $\partial_{\boldsymbol{u}}$ denotes the subdifferential of the cost function of $\text{FOP}_{\boldsymbol{\theta}}$, which generalizes the gradient to allow for non-smooth objectives. The vectors $\boldsymbol{w}_t$ and $\boldsymbol{\lambda}_t$ are dual variables. The objective function in (7a) follows the structure of Problem (6), penalizing the sum of squared duality gaps while regularizing $\boldsymbol{\theta}$ via $h(\cdot)$. Constraints (7b)-(7e) are the KKT conditions, where (7e) relaxes the strong duality condition with a duality gap $\boldsymbol{\epsilon}_t$.

Concretely, we minimize the duality gap in Problem (7) instead of the distance between given expert decisions and optimal solutions to the forward optimization problem. This aims to reduce the computational time when Problem (7) is non-convex, which will be analyzed in more detail in the following sections. Once the preliminary formulation of $FOP_\theta$ is established and the corresponding inverse optimization problem is formulated, we leverage the expert data to infer the unknown parameters within the model, with the objective of recovering a parameterization under which the observed decisions are (approximately) optimal.

### 3.3.2 SPECIAL CASE WITH THEORETICAL ANALYSIS

We next consider a special class of decision-making problems with a linear stage cost function $\boldsymbol{c}^\top \boldsymbol{u}_t$ and state-independent function $g(\cdot)$ (i.e., $B_t = B, \forall t$). We show that under this setting, our framework can achieve desirable properties such as inverse feasibility, forward stability, and computation efficiency. At the end of this section, we generalize our conclusions to a broader class of problems with quadratic stage cost functions.

We make the following remarks on this special setting. First, this setting captures a broad range of applications in resource allocation, logistics, and energy systems, where strict constraint satisfaction and fast computation are critical. Second, despite the linear cost structure, the original multi-stage decision-making problem (1) can still be challenging to solve in real-world operations due to (i) the potentially nonlinear structures of $f(\cdot)$ and $g(\cdot)$ and (ii) potentially high state and action dimensions. Therefore, a hierarchical decision-making problem is still required for computational efficiency.

In this special setting, we aim to derive an FOP formulation such that the inverse optimization is always feasible. This is important, as it can allow us to utilize any type of expert data. In this case, we formulate the estimation of $\boldsymbol{\theta}$ as

$$\min_{\boldsymbol{\theta}} \left\{ \varphi(\boldsymbol{\theta}) \mid \boldsymbol{\theta} \in \boldsymbol{\theta}_t^{\text{inv}}(\boldsymbol{u}_t) \ \forall t \in \mathcal{T}_e, \ \boldsymbol{\theta} \in \boldsymbol{\Theta} \right\}. \tag{8}$$

where $\boldsymbol{\theta}_t^{\text{inv}}(\boldsymbol{u}_t) := \left\{ \boldsymbol{\theta} \mid \boldsymbol{u}_t \in \mathcal{U}_t^{\text{opt}}(\boldsymbol{\theta}) \right\}$ is the inverse feasible set, i.e., the set of parameter values under which the expert action $\boldsymbol{u}_t$ belongs to the optimal solution set $\mathcal{U}_t^{\text{opt}}(\boldsymbol{\theta})$ of the lower-level problem $\text{FOP}_{\boldsymbol{\theta}}(\boldsymbol{x}_t, \hat{\boldsymbol{x}}_t)$. The function $\varphi(\boldsymbol{\theta})$ denotes a user-defined, application-specific objective that resolves indeterminacy by selecting among multiple feasible parameters.

As stated in Proposition 1, Problem (8) can be infeasible without a properly designed cost structure $d_{\boldsymbol{\theta}}(\cdot)$. Counterexamples are constructed in Appendix A.1.

**Proposition 1.** *The inverse feasible set for Problem (5)*

$$\boldsymbol{\theta}_t^{inv}(\boldsymbol{u}_t) := \left\{ \boldsymbol{\theta} \mid \boldsymbol{u}_t \in \mathcal{U}_t^{opt}(\boldsymbol{\theta}) \right\}.$$

*is not guaranteed to be non-empty, where $\mathcal{U}_t^{opt}(\boldsymbol{\theta})$ is defined by the KKT conditions of Problem (5) and $\boldsymbol{\theta}$ denotes unknown parameters in the problem.*

To address the sub-optimality, we propose an inverse-optimization–guided design procedure that leverages criteria such as inverse feasibility and forward stability to inform the lower-level formulation.

**Stage 1: Validating optimality of expert decisions.** We first construct a preliminary formulation of $\text{FOP}_{\boldsymbol{\theta}}$ such that the historical decisions $\{\boldsymbol{u}_t\}_{t=1}^T$ are possible to be optimal under the observed states $\{\boldsymbol{x}_t\}_{t=1}^T$. We achieve this goal by validating the feasibility of the inverse problem.

We select the terminal cost $d_{\boldsymbol{\theta}}(\cdot)$ represented by the summation of ReLU-based regularization terms, which demonstrates high efficiency despite its simplicity. Propositions 2 and 3 show the motivation to design such cost structure from both geometric and algebraic perspectives respectively.

$$\text{FOP}(\boldsymbol{x}_t, \hat{\boldsymbol{x}}_t) := \arg\min_{\boldsymbol{u}_t} \left\{ \boldsymbol{c}^\top \boldsymbol{u}_t + \sum_{k=1}^K \max\{\boldsymbol{\theta}_k^T \boldsymbol{u}_t - \nu_k, 0\} \big| \boldsymbol{a}_t + B\boldsymbol{u}_t = \hat{\boldsymbol{x}}_t, \boldsymbol{u}_t \in \widetilde{\mathcal{U}}_t \right\} \quad (9)$$

**Proposition 2.** *The inverse feasible set for Problem (9), $\boldsymbol{\theta}_t^{*inv}(\boldsymbol{u}_t) := \left\{ \boldsymbol{\theta} \mid \boldsymbol{u}_t \in \mathcal{U}_t^{*opt}(\boldsymbol{\theta}) \right\}$, is always non-empty, where $\mathcal{U}_t^{*opt}(\boldsymbol{\theta})$ is defined by KKT conditions of Problem (9).*

**Proposition 3.** *By appropriately enlarging the decision space with auxiliary variables in Problem (9), any given set of expert decisions can be made optimal in the lifted space.*

**Stage 2: Ensuring forward stability.** Although Problem (9) has proven to make expert data optimal, i.e., the inverse feasibility is guaranteed, another critical issue in inverse optimization, named forward stability, proposed by Shahmoradi & Lee (2022), is not guaranteed in the case of a linear program. The definition of forward stability is defined as follows.

**Definition 1** (Forward Instability (Shahmoradi & Lee, 2022))**.** *Given a set of expert observations $\hat{\mathcal{U}}$, the forward instability of an inverse solution $\hat{\boldsymbol{\theta}} \in \boldsymbol{\theta}^{*inv}(\hat{\mathcal{U}})$ is defined as*

$$\max_{u \in \mathcal{U}^{*opt}(\hat{\boldsymbol{\theta}})} \left\{ d(\hat{\mathcal{U}}, u) \right\}, \quad (10)$$

*where $\mathcal{U}^{*opt}(\hat{\boldsymbol{\theta}})$ denotes the set of forward optimal solutions corresponding to $\hat{\boldsymbol{\theta}}$. This value quantifies the worst-case distance between a forward solution $u$ induced by $\hat{\boldsymbol{\theta}}$ and the expert data $\hat{\mathcal{U}}$, measuring how unstable the inverse solution $\hat{\boldsymbol{\theta}}$ can be.*

To solve this issue, we include small quadratic terms into objective function to ensure strong convexity of objective function, such that forward stability is improved and the expert decisions are approximately optimal at the same time. Overall, we formulate the forward optimization problem as follows, which improves forward stability without sacrificing computational tractability at the same time.

$$\begin{aligned} \text{FOP}(\hat{\boldsymbol{x}}_t, \boldsymbol{x}_t) := \quad &\arg\min_{\boldsymbol{u}_t} \boldsymbol{c}^\top \boldsymbol{u}_t + \sum_{k=1}^K \max\{\boldsymbol{\theta}_k^T \boldsymbol{u}_t - \nu_k, 0\} + l * \|\boldsymbol{u}_t\|_2^2 \\ \text{s.t.} \quad &\boldsymbol{a}_t + B\boldsymbol{u}_t = \hat{\boldsymbol{x}}_t, \, t \in \mathcal{T}_e \\ &\boldsymbol{u}_t \in \mathcal{U}_t, \, t \in \mathcal{T}_e \end{aligned} \quad (11)$$

We propose the following inverse optimization formulation, where dual variables are denoted by $\boldsymbol{z}_{kt}, \boldsymbol{y}_{kt}, \boldsymbol{w}_t, \boldsymbol{\lambda}_t$ and $\tau_{kt}$ is the auxiliary variable. $g(\boldsymbol{\lambda}_t, \boldsymbol{z}_t, \boldsymbol{y}_t, \boldsymbol{w}_t)$ is the dual objective function with explicit form in this problem. To keep the formulation concise, we omit the explicit form of this function.

$$\min_{\boldsymbol{z}_{kt}, \boldsymbol{y}_{kt}, \boldsymbol{\lambda}_t, \boldsymbol{w}_t, \boldsymbol{\epsilon}_t, \boldsymbol{\theta}^{(k)}} \sum_{t \in \mathcal{T}_e} \|\boldsymbol{\epsilon}_t\|_2^2 + \rho \sum_k \|\boldsymbol{\theta}^{(k)}\|_2^2 \quad (12a)$$

$$\text{s.t.} \quad 1 - \boldsymbol{z}_{kt} - \boldsymbol{y}_{kt} = 0, \, k = 1, ...K, \, t \in \mathcal{T}_e \quad (12b)$$

$$\boldsymbol{c}^T + \boldsymbol{\lambda}_t^T H + \boldsymbol{w}_t^T B + \sum_k \boldsymbol{z}_{kt} \boldsymbol{\theta}_k^T + 2l\boldsymbol{u}_t = 0, \, t \in \mathcal{T}_e \quad (12c)$$

$$g(\boldsymbol{\lambda}_t, \boldsymbol{z}_t, \boldsymbol{y}_t, \boldsymbol{w}_t) + \boldsymbol{\epsilon}_t = \boldsymbol{c}^\top \boldsymbol{u}_t + \sum_{k=1}^K \tau_{kt} + l\boldsymbol{u}_t^T \boldsymbol{u}_t, \, t \in \mathcal{T}_e \quad (12d)$$

$$\tau_{kt} \geq \boldsymbol{\theta}_k^T \boldsymbol{u}_t - \nu_k, \, k = 1, ...K, \, t \in \mathcal{T}_e \quad (12e)$$

$$\boldsymbol{z}_{kt}, \boldsymbol{y}_{kt}, \boldsymbol{\lambda}_t, \tau_{kt} \geq 0, \, k = 1, ...K, \, t \in \mathcal{T}_e \quad (12f)$$

We minimize duality gap together with regularizing norm of $\boldsymbol{\theta}_k$ for robustness. This reformulation aims to reduce the computational time required to solve the non-convex problem which poses

significant challenges for computational tractability. Instead of minimizing distance between optimal solutions and given expert decisions, this formulation reduces the number of bilinear terms by avoiding introducing new variables corresponding to the optimal solutions of the forward problem 11. What's more, quadratic term is introduced to ensure that forward stability is improved. Proposition 4 demonstrates that the distance between the input and optimal solutions remains bounded. By introducing a sufficient number of ReLU-based terms, we can constrain the values of $\epsilon_0$ within a small range, thereby controlling the upper bound of the distance.

**Proposition 4.** *Problem (12) yields optimal solutions $\boldsymbol{u}_t^*$ satisfying $|f(\boldsymbol{u}_t^*) - f(\boldsymbol{u}_t)| \leq \epsilon_0$, s.t.*

$$\|\boldsymbol{u}_t^* - \boldsymbol{u}_t\| \leq \sqrt{\frac{\epsilon_0}{l}} \tag{13}$$

*where $\epsilon_0 = \max_t \boldsymbol{\epsilon}_t^*$ and $\boldsymbol{u}_t$ is the given expert decision in time step $t$.*

For solving Problem (12), it can be reformulated as a Mixed Integer Program and solved using the spatial branch-and-bound algorithm within a reasonable time (Smith & Pantelides, 1999). By minimizing the duality gap instead of the distance between optimal solutions and given expert solutions, the number of bilinear terms is reduced from $\mathcal{O}(|\mathcal{A}|K + |\mathcal{A}|^2)$ to $\mathcal{O}(|\mathcal{A}|K)$. Thus, the number of constraints introduced by the construction of the McCormick envelope can be significantly reduced.

For the broader class of control problems with quadratic objective functions, we can still infer parameters as shown in Problem (12). Proposition 4 can be extended as shown below.

**Corollary 1.** *When the objective function is given by $\boldsymbol{u}^T R \boldsymbol{u} + \boldsymbol{x}^T Q \boldsymbol{x}$ with the assumption $R \succ 0$, the upper bound will be achieved by $\sqrt{\frac{\lambda_{min}(R)}{l}}$, where $\lambda_{min}(R)$ denotes the minimum eigenvalue of $R$.*

## 4 CASE STUDY

In this section, we evaluate the proposed framework through three case studies from different fields: (1) autonomous vehicle rebalancing; (2) supply chain inventory management problems, and (3) mobile robot navigation. The results highlight its ability to enhance decision quality and provide interpretable solutions in dynamic environments. The details of the three environments are given in Appendix B.

### 4.1 EXPERIMENTAL SETUP

**Data Collection:** We generate expert demonstrations by simulating a full-horizon MPC controller over a finite horizon $T$. The dataset consists of state-action pairs collected offline, designed to cover diverse operating conditions for each distinct task. Crucially, we utilize only a handful of these demonstrations to infer the formulation of the lower-level optimization policy via inverse optimization

**RL Implementation:** We employ A2C with Graph Neural Networks for the AV and Supply Chain tasks, and Soft Actor-Critic (SAC) for the Mobile Robot task. Detailed network architectures and hyperparameters are provided in Appendix C.1.

**Baselines:** We compare our results to:

1. MPC: MPC that receives perfect information of the system dynamics and future evolution of task-specific environmental variables.

2. Bi-level-unchanged: Bi-level framework where lower-level optimization policy is unchanged with original single-step cost (Gammelli et al., 2023).

3. Bi-level-cvxpy: Embed optimization problem as a differential layer and solve for the optimal parameters using gradient-based method (Agrawal et al., 2019).

4. End-to-end RL: End-to-end reinforcement learning that trains a single model to directly map inputs to the final control actions.

5. Learning-Based Terminal Cost Approximation: Assume quadratic structure of terminal cost and estimate the value function (Abdufattokhov et al., 2021).

## 4.2 MODEL EVALUATION

We compare the bi-level framework equipped with a learned lower-level policy against variant baselines. Table 1 and 2 present the system performance. Our Bi-level-learned approach improves upon the Bi-level-unchanged and Bi-level-cvxpy baseline by employing a refined lower-level formulation. And our bi-level framework significantly outperforms end-to-end architectures, revealing a substantial performance gap.

Moreover, compared with MPC, the bi-level framework shortens the planning horizon from T steps to one step, which significantly reduces decision-making time. We compare the time spent to make decisions for one step in Table 3. It shows a significant reduction in runtime which indicates that our method can significantly improve computational efficiency without substantially compromising solution quality, thereby enabling faster real-time decision-making.

These results highlight the effectiveness of optimizing the lower-level problem structure in enhancing overall system performance. Besides, scalability analysis, sensitivity analysis, interpretations of the results and discussions of methods are provided and discussed in the Appendix C.

Table 1: Performance Comparison on Autonomous Vehicles Rebalancing and Supply Chain Inventory Management.

| Method | Autonomous Vehicle Rebalancing | | Inventory Management | |
| --- | --- | --- | --- | --- |
| | Reward | Served Demand | Reward | Served Demand |
| MPC | 35725 ($\pm$41.6) | 3203 ($\pm$3.07) | 11335 ($\pm$34.3) | 1337 ($\pm$5.61) |
| End-to-end RL | 20989 ($\pm$213.5) | 2334.8 ($\pm$14.9) | 2764 ($\pm$90.9) | 476 ($\pm$18.4) |
| Bi-level-unchanged | 33406 ($\pm$128.1) | 2993 ($\pm$8.06) | 7491 ($\pm$26.5) | 778 ($\pm$2.62) |
| Bi-level-cvxpy | 34403 ($\pm$108.7) | 3086 ($\pm$7.06) | 9268 ($\pm$32.1) | 912 ($\pm$2.20) |
| Bi-level-learned (Ours) | **35019** ($\pm$53.0) | **3141** ($\pm$6.14) | **9442** ($\pm$42.3) | **930** ($\pm$1.17) |

Table 2: Performance Comparison on Mobile Robot Navigation.

| Method | Travel Time (s) | Path Length (m) | Energy (J) |
| --- | --- | --- | --- |
| MPC | 3.50 | 3.50 | 4.81 |
| End-to-end RL | 7.60 $\pm$ 0.11 | 4.29 $\pm$ 0.02 | 6.78 $\pm$ 0.23 |
| Bi-level-unchanged | 9.14 $\pm$ 0.08 | 4.35 $\pm$ 0.02 | 6.20 $\pm$ 0.13 |
| Bi-level-learned (Ours) | **4.82** $\pm$ 0.30 | **4.16** $\pm$ 0.19 | **2.92** $\pm$ 1.04 |
| Bi-level-cvxpy | * | * | * |

An asterisk (*) indicates that no valid result was obtained for this method.

Table 3: Comparison of Computational Time (in seconds) Across Various Scenarios

| Scenarios | Our Method | MPC |
| --- | --- | --- |
| AV Rebalancing | **0.025** | 1.44 |
| SCIM | **0.034** | 0.082 |
| Mobile Robot | **0.0187** | 0.0275 |

## 5 FUTURE WORK

There are several promising directions for future research. First, to enhance the generalizability of the proposed framework, it's critical to make the learning procedure more generalizable. We aim to extend it to settings involving more complex system dynamics beyond the current control-affine formulation to figure out how the current framework can be adapted to different problems more efficiently. Second, as the inverse optimization component can become computationally intensive in large-scale scenarios, developing more scalable and efficient algorithms for inverse problem solving

remains an important avenue. Besides, the subgoal in the hierarchical framework may not be retrived from expert demonstrations which requires further analysis to extend the applicability of the proposed method.

## 6 CONCLUSION

In this work, we propose a hierarchical reinforcement learning and optimization framework that addresses key challenges in real-time, safety-critical decision-making scenarios. By leveraging intrinsic motivation, the upper-level policy generates subgoals that abstract planning objectives, while the lower-level controller executes these subgoals through a structured optimization problem. A central contribution of this work lies in the data-driven design of the lower-level optimization formulation based on expert demonstrations. Our method is evaluated on several real-world scenarios from different fields, where it demonstrates strong empirical performance. Furthermore, our model exhibits improved alignment with expert decisions and offers interpretable, structured control policies. This work highlights promising directions for combining learning-based goal abstraction with structured optimization. Our results suggest that such structured, bi-level approaches are promising for scaling decision-making in dynamic and safety-critical domains.

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

# A PROOF OF PROPOSITIONS

## A.1 PROPOSITION 1

The inverse feasible set for Problem (4)

$$\boldsymbol{\theta}_t^{\text{inv}}(\boldsymbol{u}_t) := \left\{ \boldsymbol{\theta} \mid \boldsymbol{u}_t \in \mathcal{U}_t^{\text{opt}}(\boldsymbol{\theta}) \right\}.$$

is not guaranteed to be non-empty, where $\mathcal{U}_t^{\text{opt}}(\boldsymbol{\theta})$ is defined by KKT optimality conditions of Problem (4).

*Proof.* We explore whether there exists $\boldsymbol{\theta}$ belonging to the inverse feasible set for Problem (4), which is making $\boldsymbol{u}_t$ to satisfy all KKT conditions.

- KKT conditions for Problem (4):

    1. Stationary Conditions:
    $$c^T + \boldsymbol{\lambda}_t^T H + \boldsymbol{w}_t^T B = 0$$

    2. Complementary Slackness:
    $$\boldsymbol{\lambda}_t^T (H\boldsymbol{u}_t - h) = 0$$

    3. Dual Feasibility and Primal Feasibility:
    $$\boldsymbol{\lambda}_t \geq 0$$
    $$H\boldsymbol{u}_t \leq h$$

We assume $H\boldsymbol{u}_t \leq h$ is not active, which means $\boldsymbol{\lambda}_t = 0$. The system $B^T \boldsymbol{w}_t = -c$ has no feasible solution if the vector $-c$ does not lie in the column space of $B^T$. For example, when the system is overdetermined and $rank(B^T)$ is larger than dimension of $\boldsymbol{w}_t$. $\qquad\square$

## A.2 PROPOSITION 2

The inverse feasible set for Problem (9)

$$\boldsymbol{\theta}_t^{*\text{inv}}(\boldsymbol{u}_t) := \left\{ \boldsymbol{\theta} \mid \boldsymbol{u}_t \in \mathcal{U}_t^{*\text{opt}}(\boldsymbol{\theta}) \right\}.$$

is always non-empty, where $\mathcal{U}_t^{*\text{opt}}(\boldsymbol{\theta})$ is defined by KKT optimality conditions of Problem (9).

*Proof.* We explore whether there exists $\boldsymbol{\theta}$ belonging to the inverse feasible set for Problem (9), which is making $\boldsymbol{u}_t$ to satisfy all KKT conditions.

- KKT conditions for Problem 9:

    1. Stationary Conditions:

    $$c^T + \sum_{k=1}^{K} \boldsymbol{z}_{kt} \boldsymbol{\theta}_k^T + \boldsymbol{\lambda}_t^T H + \boldsymbol{w}_t^T B = 0, \quad for\, t \in \mathcal{T}_e$$
    $$1 - \boldsymbol{z}_{kt} - \boldsymbol{y}_{kt} = 0, \quad for\, k = 1, ...K\, and\, t \in \mathcal{T}_e$$

    2. Complementary Slackness:

    $$\boldsymbol{z}_{kt}(\tau_{kt} - \boldsymbol{\theta}_k^T \boldsymbol{u}_t + \nu_k) = 0, \quad for\, k = 1, ...K\, and\, t \in \mathcal{T}_e$$
    $$\boldsymbol{y}_{kt} \tau_{kt} = 0, \quad for\, k = 1, ...K\, and\, t \in \mathcal{T}_e$$
    $$\boldsymbol{\lambda}_t^T (H\boldsymbol{u}_t - h) = 0, \quad for\, t \in \mathcal{T}_e$$

    3. Dual Feasibility and Primal Feasibility:

    $$\boldsymbol{z}_{kt},\, \boldsymbol{y}_{kt},\, \boldsymbol{\lambda}_t \geq 0, \quad for\, k = 1, ...K\, and\, t \in \mathcal{T}_e$$
    $$H\boldsymbol{u}_t \leq h, \quad for\, t \in \mathcal{T}_e$$
    $$\tau_{kt} \geq \boldsymbol{\theta}_k^T \boldsymbol{u}_t - \nu_k, \tau_{kt} \geq 0, \quad for\, k = 1, ...K\, and\, t \in \mathcal{T}_e$$

There exist trivial feasible solutions to satisfy all KKT conditions:

We assume $\boldsymbol{z}_{kt} = 1$, $\boldsymbol{y}_{kt} = 0$ and $\boldsymbol{\lambda}_t = 0$. Then the set of KKT conditions are transformed into

$$c^T + \sum_{k=1}^{K} \boldsymbol{\theta}_k^T + \boldsymbol{w}_t^T B = 0, \quad for \ t \in \mathcal{T}_e \tag{14}$$

$$\tau_{kt} = \boldsymbol{\theta}_k^T \boldsymbol{u}_t - \nu_k, \ \tau_{kt} \geq 0, \quad for \ k = 1, ...K \ and \ t \in \mathcal{T}_e$$

We must find $\{\boldsymbol{\theta}_k\}_{k=1}^K$ to satisfy $c^T + \sum_{k=1}^{K} \boldsymbol{\theta}_k^T + \boldsymbol{w}_t^T B = 0$ and let $\nu_k = \min_{t=1}^T \{\boldsymbol{\theta}_k^T \boldsymbol{u}_t\}$ □

### A.3 PROPOSITION 3

By appropriately enlarging the decision space with auxiliary variables in Problem (9), any given set of expert decisions can be made optimal in the lifted space.

*Proof.* Consider a collection of expert decisions $\{u_i\}_{i=1}^N$ with $u_i \in \mathbb{R}^n$. Problem (9) can be reformulated as a linear program by introducing auxiliary variables $\{t_k\}_{k=1}^K$, which effectively enlarges the decision space from $\mathbb{R}^n$ to $\mathbb{R}^{n+K}$. For clarity, let us focus on the case $K = 1$. By adding the constraints

$$t_k \geq \boldsymbol{\theta}_k^\top u_i - \nu, \tag{15}$$

We can at least guarantee that all expert decisions $\{u_i\}_{i=1}^N$ lie on the same $n$-dimensional face of the feasible polytope in the augmented space by making the new added constraint active. This face can be made parallel to the hyperplane defined by the objective function by adjusting the new added constraint.

This situation, however, represents an extreme case. In practice, the set $\{u_i\}_{i=1}^N$ may lie within a face of dimension strictly smaller than $n$. To illustrate, suppose $u_i \in \mathbb{R}^2$. If all expert decisions lie on a one-dimensional face (an edge) of the feasible polytope, then they remain on a corresponding one-dimensional face in the lifted $\mathbb{R}^3$ space. Assume that this edge arises as the intersection of two adjacent facets, whose supporting hyperplanes have normal vectors $\vec{n}_1$ and $\vec{n}_2$, respectively. The direction vector of the edge is then given by

$$\vec{d}_{\text{edge}} = \vec{n}_1 \times \vec{n}_2, \tag{16}$$

since it is simultaneously orthogonal to both $\vec{n}_1$ and $\vec{n}_2$.

For the expert decisions to be optimal, it suffices that the objective hyperplane be parallel to this edge. Equivalently, the edge direction $\vec{d}_{\text{edge}}$ must be orthogonal to the objective normal $\vec{n}_{\text{obj}}$, i.e.,

$$\vec{d}_{\text{edge}} \cdot \vec{n}_{\text{obj}} = 0. \tag{17}$$

This condition establishes the required relationship between the optimal geometry and the objective orientation. Thus, we can solve the equation for feasible unknown parameters in the terminal cost. □

### A.4 PROPOSITION 4

Problem (12) yields optimal solutions $\boldsymbol{u}_t^*$ satisfying $|f(\boldsymbol{u}_t^*) - f(\boldsymbol{u}_t)| \leq \epsilon_0$, s.t.

$$\|\boldsymbol{u}_t^* - \boldsymbol{u}_t\| \leq \sqrt{\frac{\epsilon_0}{l}} \tag{18}$$

*Proof.* First, we show that the feasible region of Problem (12) is non-empty. Since the strong duality constraint has been relaxed, the main task reduces to verifying the stationarity condition

$$c^T + \boldsymbol{\lambda}_t^T H + \boldsymbol{w}_t^T B + \sum_k \boldsymbol{z}_{kt} \boldsymbol{\theta}_k^T + 2l\boldsymbol{u}_t = 0, \qquad t = 1, \dots, T. \tag{19}$$

For the $i$-th expert decision, suppose there are $n_i$ active inequality constraints. Each such active constraint introduces one additional degree of freedom in the stationarity condition. Together with the $N$ equality constraints, the total number of effective constraints across $T$ time periods is

$$\sum_{i=1}^{T} n_i + NT. \tag{20}$$

Since the decision variable $c$ lies in $\mathbb{R}^M$, the stationarity condition can be satisfied provided that the number of auxiliary terms is large enough to compensate for the remaining degrees of freedom. More precisely, the required number of ReLU terms is bounded above by

$$\max\left\{0,\ T - \frac{\sum_{i=1}^{T} n_i + NT}{M}\right\}. \tag{21}$$

This establishes the feasibility of the relaxed problem. Next, we derive the upper bound of distance.

Based on strong convexity of $f(u)$, we have

$$f(\boldsymbol{u}_t) \geq f(\boldsymbol{u}_t^*) + \nabla f(\boldsymbol{u}_t^*)^\top (\boldsymbol{u}_t - \boldsymbol{u}_t^*) + \frac{\mu}{2}\|\boldsymbol{u}_t - \boldsymbol{u}_t^*\|^2 \tag{22}$$

where $\nabla^2 f(\boldsymbol{u}_t) \geq \mu I > 0$. By stationary conditioin $\nabla f(\boldsymbol{u}_t^*) = 0$,

$$f(\boldsymbol{u}_t) - f(\boldsymbol{u}_t^*) \geq \frac{\mu}{2}\|\boldsymbol{u}_t - \boldsymbol{u}_t^*\|^2 \tag{23}$$

The dual objective function $g(\boldsymbol{\lambda}_t, \boldsymbol{z}_t, \boldsymbol{y}_t, \boldsymbol{w}_t) \leq f(\boldsymbol{u}_t^*) \leq f(\boldsymbol{u}_t)$ and we have $f(\boldsymbol{u}_t) - g(\boldsymbol{\lambda}_t, \boldsymbol{z}_t, \boldsymbol{y}_t, \boldsymbol{w}_t) \leq \epsilon_0$ by constraining strong duality gap, s.t.

$$f(\boldsymbol{u}_t) - f(\boldsymbol{u}_t^*) \leq \epsilon_0 \tag{24}$$

Thus, we have

$$\|\boldsymbol{u}_t - \boldsymbol{u}_t^*\|^2 \leq \epsilon_0 * \frac{2}{\mu} = \frac{\epsilon_0}{l} \tag{25}$$

$\square$

## B DETAILS OF CASE STUDIES

We consider three simulation scenarios described as follows:

**Autonomous Vehicle Rebalancing.** Autonomous Mobility-on-Demand (AMoD) systems are an evolving mode of transportation in which a centrally coordinated fleet of self-driving vehicles dynamically serves travel requests. In real-world systems, the effectiveness of rebalancing strategies is central to the overall system performance, with sub-optimal strategies potentially exacerbating congestion through unnecessary trips or increased passenger waiting time. The control of these systems is typically formulated as a large network optimization problem. This framework comprises two stages: (1) determining the desired distribution of idle vehicles $\hat{q}_t$ through the use of the learned policy $\pi_\phi(\hat{q}_t|s_t)$ by reinforcement learning, (2) converting this distribution to a passenger flow $g_{ij}^t$ and rebalancing flow $f_{ij}^t$ by solving a linear control problem.

**Supply Chain Inventory Management.** In the supply chain inventory management task, we aim to determine the optimal ordering and distribution strategies across a network of interconnected warehouses and retail stores to satisfy customer demand while minimizing storage, production and transportation costs. We choose upper-level goals as (i) desired production in warehouse nodes $\hat{w}_i^t$ and (ii) desired inventory in store nodes $\hat{q}_i^t$. And the lower-level optimization module determines the amount of commodities $\boldsymbol{w}_i^t$ to order in each warehouse, and the shipping flow $f_{ij}^t$ from warehouses to stores.

**Mobile Robot Navigation.** In the mobile robot navigation task, a ground robot moves from a start location to a target destination while avoiding static obstacles. The control problem is typically formulated in a hierarchical framework. At the high level, a learned policy $\pi_{\boldsymbol{\theta}}(\hat{p}_t \mid s_t)$ outputs intermediate waypoints, conditioned on the robot's state, the goal position, and obstacle information. At the low level, these actions are converted into executable control inputs $(v_t, \boldsymbol{\omega}_t)$ for the unicycle dynamics, ensuring feasibility under kinematic and dynamic constraints.

The detailed formulations of the lower-level optimization problems for these three environments are given as follows.

## B.1 AUTONOMOUS VEHICLES REBALANCING

The lower-level optimization policy is represented in a matrix form as follows.

$$\min_{f_t, g_t} c^T f_t - p^T g_t + \sum_k \max\{-\boldsymbol{\theta}_f^{(k)^T} f_t + \boldsymbol{\theta}_g^{(k)^T} g_t - \mu^{(k)}, 0\}$$

$$s.t. \quad q_t - A(f_t + g_t) \geq \hat{q}_t$$
$$q_t - G(f_t + g_t) \geq 0 \tag{26}$$
$$f_t \geq 0, g_t \geq 0$$
$$g_t \leq \boldsymbol{\lambda}_t$$

where A is the instance matrix, $\boldsymbol{\lambda}_t$ is the travel requests in time slot $t$ and $f_t, g_t$ denote rebalancing flow and passenger flow respectively.

**Proposition 5.** *The inverse feasible set for Problem 26*

$$\boldsymbol{\theta}_t^{*inv}(\mathbf{f}_t, \mathbf{g}_t) := \left\{ \boldsymbol{\theta} \mid \mathbf{f}_t, \mathbf{g}_t \in \mathcal{X}_t^{*opt}(\boldsymbol{\theta}) \right\}. \tag{27}$$

*is always non-empty, where $\mathcal{X}_t^{*opt}(\boldsymbol{\theta})$ is defined by KKT optimality conditions of Problem 26.*

Next, we will prove this proposition and also show that the solved parameters have practical significance which has not been discussed further in the general setting.

*Proof.* The inverse optimization solving for unknown parameters in Problem 26 is formulated as follows

$$\min_{\boldsymbol{x}_t, \boldsymbol{y}_t, \boldsymbol{w}_t, v_{1t}, v_{2t}, \boldsymbol{z}_{1t}, \boldsymbol{z}_{2t}, \boldsymbol{\tau}_t, \boldsymbol{\theta}, \mu} \sum_k \|(\boldsymbol{\theta}^{(k)})\|^2$$

$$1 - \boldsymbol{z}_{1t}^{(k)} - \boldsymbol{z}_{2t}^{(k)} = 0, \ k = 1, ...K, \ t = 1, ...T$$

$$c^T + \boldsymbol{x}_t^T A + \boldsymbol{y}_t^T G - \boldsymbol{w}_t^T - \sum_k \boldsymbol{z}_{1t}^{(k)} \boldsymbol{\theta}_f^{(k)^T} + 2l_1 f_t = 0, \ t = 1, ...T$$

$$-p^T + \boldsymbol{x}_t^T A + \boldsymbol{y}_t^T G - v_{1t}^T + v_{2t}^T + \sum_k \boldsymbol{z}_{1t}^{(k)} \boldsymbol{\theta}_g^{(k)^T} + 2l_2 g_t = 0, \ t = 1, ...T \tag{28}$$

$$\boldsymbol{x}_t^T (q_t - A(f_t + g_t) - \hat{q}_t) = 0, \ t = 1, ...T$$

$$\boldsymbol{y}_t^T (q_t - G(f_t + g + t)) = 0, \ t = 1, ...T$$

$$\boldsymbol{w}_t^T f_t = 0, \ t = 1, ...T$$

$$v_{1t}^T g_t = 0, \ t = 1, ...T$$

$$v_{2t}^T (g_t - \boldsymbol{\lambda}_t) = 0, \ t = 1, ...T$$

$$\text{(other feasiblity constraints)}$$

Given historical data $\{f_i, g_i, q_i, \lambda_i\}_{i=1}^N$, we consider the worst-case scenario where

$$Gf_i + Gg_i \neq q_i, \quad f_i \neq 0, \quad g_i \neq 0, \quad g_i \neq \lambda_i \quad \forall i = 1, \dots, N.$$

This implies that all corresponding dual variables

$$\boldsymbol{y}_i = \boldsymbol{w}_i = v_{1i} = v_{2i} = 0 \quad \forall i = 1, \dots, N.$$

We proceed with the analysis under this assumption. Moreover, We further set

$$\boldsymbol{z}_{1i}^{(k)} = 1, \ \boldsymbol{z}_{2i}^{(k)} = 0$$

which simplifies inverse problem corresponding to Problem 26 into the following form:

$$\min_{\boldsymbol{\theta}^{(k)}, \mu^{(k)}, x_i} \quad \sum_k \|\boldsymbol{\theta}^{(k)}\|^2$$

$$\text{s.t.} \quad -\boldsymbol{\theta}_f^{(k)\top} f_i + \boldsymbol{\theta}_g^{(k)\top} g_i - \mu^{(k)} = \tau_i^{(k)}, \quad \forall k = 1, \dots, K,$$

$$c^\top + x_i^\top A - \sum_k \boldsymbol{\theta}_f^{(k)\top} = 0, \tag{29}$$

$$-p^\top + x_i^\top A + \sum_k \boldsymbol{\theta}_g^{(k)\top} = 0,$$

$$\text{(other feasibility constraints).}$$

Now assume $x_i \preceq \epsilon \mathbf{1}$ for some small $\epsilon > 0$. Then we have

$$-\epsilon \mathbf{1} \preceq x_i^\top A \preceq \epsilon \mathbf{1}.$$

Suppose $\mathbf{c}, \mathbf{p} \in \mathbb{R}_{++}^n$. We can select $\epsilon$ such that

$$\min_{1 \leq j \leq M} c_j \geq \epsilon, \qquad \min_{1 \leq j \leq M} p_j \geq \epsilon.$$

Then the constraints in 29 reduce to

$$-\boldsymbol{\theta}_f^{(k)\top} f_i + \boldsymbol{\theta}_g^{(k)\top} g_i - \mu^{(k)} \geq 0, \quad \forall k = 1, \dots, K,$$

$$c^\top + x_i^\top A - \sum_k \boldsymbol{\theta}_f^{(k)\top} = 0, \tag{30}$$

$$-p^\top + x_i^\top A + \sum_k \boldsymbol{\theta}_g^{(k)\top} = 0.$$

Assume further that $K = 1$ (for simplicity). Then the last two equalities yield:

$$\boldsymbol{\theta}_f = c + A^\top x_i, \qquad \boldsymbol{\theta}_g = p - A^\top x_i.$$

In the context of autonomous vehicle rebalancing, it is reasonable to assume that the total revenue exceeds the cost, i.e., $p^\top g_i > c^\top f_i$. Then we compute:

$$
\begin{aligned}
-\boldsymbol{\theta}_f^\top f_i + \boldsymbol{\theta}_g^\top g_i - \mu &= -(c^\top + x_i^\top A) f_i + (p^\top - x_i^\top A) g_i - \mu \\
&= (p^\top g_i - c^\top f_i) - x_i^\top A (f_i + g_i) - \mu \\
&\geq (p^\top g_i - c^\top f_i) - \epsilon \mathbf{1}^\top (f_i + g_i) - \mu.
\end{aligned}
\tag{31}
$$

Therefore, we can choose $\epsilon$ sufficiently small and $\mu$ accordingly, such that the right-hand side of 31 remains non-negative. This guarantees that the constraints in 30 are satisfied, hence the problem admits a feasible solution. $\qquad\square$

Problem 26 is constructed so that the historical expert decisions $\{f_i, g_i\}_{i=1}^N$ are optimal solutions to the forward problem. There exist model parameters $\{\boldsymbol{\theta}_f^{(k)}, \boldsymbol{\theta}_g^{(k)}, \mu^{(k)}\}_{k=1}^K$ and desired next states $\{\hat{q}_i\}_{i=1}^N$, which can be learned via reinforcement learning, such that the expert decisions become approximately optimal.

## B.2 Supply Chain Inventory Management

The lower-level policy is formulated as follows

$$\min_{f_{ij}^t, \boldsymbol{w}_i^t} \quad \sum_{i \in V_W} m_i^O \cdot \boldsymbol{w}_i^t + \sum_{(i,j) \in \mathcal{E}} m_{ij}^T \cdot f_{ij}^t + \sum_{i \in V_S} |\epsilon_i^f| + \sum_{i \in V_W} |\epsilon_i^w| + \sum_{k=1}^K \max\{\boldsymbol{\theta}_k^T f - \mu_k, 0\} \tag{32a}$$

$$\text{s.t.} \quad \sum_{j \in N^-(i)} f_{ji}^t + \epsilon_i^f = \hat{q}_i^{t+1}, i \in V_S \tag{32b}$$

$$q_i^t + \sum_{j \in N^-(i)} f_{ji}^t - d_i^t \le c_i^t, i \in V_S \tag{32c}$$

$$\sum_{j \in N^+(i)} f_{ij}^t \le q_i^t, i \in V_W \tag{32d}$$

$$q_i^t + \boldsymbol{w}_i^t - \sum_{j \in N^+(i)} f_{ij}^t \le c_i^t, i \in V_W \tag{32e}$$

$$\boldsymbol{w}_i^t + \epsilon_i^w = \hat{w}_i^t, i \in V_W \tag{32f}$$

$$f_{ij}^t \ge 0, (i,j) \in \mathcal{E}, \boldsymbol{w}_i^t \ge 0, i \in V_W \tag{32g}$$

where $m^o, m^T, d, c$ are production cost, transportation cost, customer demand and capacity respectively, $\boldsymbol{w}_i$ denotes order quantity in warehouse $i$ and $f_{ij}$ denotes the departing flow from warehouse $i$ to store $j$, and $q$ denotes the inventory level.

## B.3 Mobile Robot

$$\min_{\boldsymbol{u}_t = (v_t, \boldsymbol{\omega}_t)} \quad c_p \|p_{t+1} - \hat{p}_t\|^2 + c_h \|\boldsymbol{\theta}_{t+1} - \hat{\boldsymbol{\theta}}_t\|^2 + c_u \|\boldsymbol{u}_t\|^2 + \sum_{k=1}^K \max\{\alpha_k \boldsymbol{u}_t - \beta_k, 0\} \tag{33a}$$

$$\text{s.t.} \quad x_{t+1} = \boldsymbol{x}_t + T_s v_t \cos \boldsymbol{\theta}_t \tag{33b}$$

$$\boldsymbol{y}_{t+1} = \boldsymbol{y}_t + T_s v_t \sin \boldsymbol{\theta}_t \tag{33c}$$

$$\boldsymbol{\theta}_{t+1} = \boldsymbol{\theta}_t + T_s \boldsymbol{\omega}_t \tag{33d}$$

$$v_{\min} \le v_t \le v_{\max} \tag{33e}$$

$$\omega_{\min} \le \boldsymbol{\omega}_t \le \omega_{\max} \tag{33f}$$

$$\|p_{t+1} - p_j^{\text{obs}}\| \ge r_j + d_{\text{safe}}, \quad \forall j = 1, \dots, M \tag{33g}$$

where $v_t$ and $\boldsymbol{\omega}_t$ denote speed and angular speed of mobile robot. $p_t = (\boldsymbol{x}_t, \boldsymbol{y}_t)$ represents its position and $\hat{p}_t$ is the reference waypoint generated by upper-level RL module. The objective is to reach the final goal and avoid obstacles.

## C Further Experimental Results and Discussions

### C.1 RL Methodology and Experimental Setup

- **Autonomous Vehicle Rebalancing**: For the Autonomous Vehicle Rebalancing experiment, we train an A2C-GNN agent as in (Gammelli et al., 2023) using on-policy updates with a discount factor of 0.99, a maximum of 50 decision steps per episode, and 10000 training episodes. The actor and critic share the same EdgeConv-based GNN encoder and each head is implemented as a two-layer multilayer perceptron with 256 hidden units, and both networks are optimized with the Adam optimizer with a learning rate of $5 \times 10^{-5}$. At the end of each episode we normalize the discounted returns, perform a single A2C update of actor and critic parameters with gradient-norm clipping at 5 to improve stability, and we save the checkpoint achieving the highest cumulative training reward for later evaluation.

- **Inventory Management**: We adopt a graph neural network-based Advantage Actor–Critic (A2C) method as in (Gammelli et al., 2023) to learn the control policy. Training is performed with an on-policy A2C update, using a discount factor of 0.99, a maximum episode length of 30 steps, and up to 20,000 training episodes. Both actor and critic are optimized with Adam with learning rate 5e-5, and gradient norm clipping is applied to improve training stability.

- **Mobile Robot Navigation**: For the mobile robot, we use an off-policy Soft Actor–Critic (SAC) algorithm. SAC is trained with a replay buffer of size $10^6$, a mini-batch size of 256, Adam optimizer with learning rate $3 \times 10^{-4}$ for both actor and critic, discount factor $\gamma = 0.99$, soft target network updates with $\tau = 0.005$, and automatic entropy tuning with target entropy set to $-\dim(\mathcal{A})$. Gradient updates start once the replay buffer contains more than 400 transitions, and the agent is trained for 500 episodes, periodically saving the actor network during training.

## C.2 MOTIVATIONS FOR INVERSE OPTIMIZATION APPROACH

We justify our choice of the proposed inverse optimization framework from two perspectives: the structural design of the cost function and the efficacy of the parameter estimation.

### C.2.1 STRUCTURAL DESIGN GUIDED BY THEORETICAL PROPERTIES

The primary motivation for adopting the inverse optimization framework lies in its ability to theoretically inform the structure of the lower-level cost function. Standard approaches often heuristically assume a fixed cost form (e.g., quadratic). We implement the method proposed by Abdufattokhov et al. (2021) which assumes quadratic structure of terminal cost. We compare the performance on autonomous vehicle rebalancing task. As table 4 suggests, there exists a large optimality gap.

Table 4: Performance Comparison on Autonomous Vehicle (AV) Rebalancing Task.

| Scenario | Method | Performance Metrics | |
|---|---|---|---|
| | | **Reward** | **Served Demand** |
| Scenario 1 | MPC | 35725 ($\pm$41.6) | 3203 ($\pm$3.07) |
| | Bi-level-learned (Ours) | **35019** ($\pm$53.0) | **3141** ($\pm$6.14) |
| | Value Approximation | 24774 | 2395 |
| Scenario 2 | MPC | 68637 ($\pm$194) | 7540 ($\pm$14.1) |
| | Bi-level-learned (Ours) | **61880** ($\pm$370.3) | **6619** ($\pm$22.3) |
| | Value Approximation | 29905 | 3160 |

### C.2.2 PARAMETER ESTIMATION EFFICACY

Beyond the cost formulation itself, we evaluate Inverse Optimization as a solver against two alternative learning paradigms. Under the assumption that RL enable to offer optimal upper-level goals, we depart the learning of cost structure from the bi-level framework training and explore which method mimics the expert decisions best.

Table 5: Similarity scores between expert decisions and decisions chosen by different methods.

| | AV Rebalancing | | Inventory Management |
|---|---|---|---|
| **Method** | **Reb. flow** | **Pax. flow** | **Commodity. flow** |
| Value Approximation | 0.522 | 0.943 | 0.461 |
| cvxpylayers | 0.541 | 0.946 | 0.445 |
| Inverse Optimization | **0.703** | **0.958** | **0.997** |

Based on Table 5, inverse optimization outperforms the other two methods to large extent. First, value function approximation method doesn't perform well in both tasks since it cannot embed the

designed structure of lower-level policy explicitly. Second, cvxpylayers perform well on AV Rebalancing task in certain scenario, but fail in some other tasks. It's due to the following reasons. First, such gradient-based optimization method is sensitive to the differentiation. When the derivative of objective with respect to the unknown parameter is small, the algorithm will struggle to optimize the parameters. Second, the optimal distribution of parameters may be complex, and thus challenging to estimate within the scope of commonly distribution families.

## C.3 INTERPRETABILITY

**Interpretations of Optimal Parameters.** The optimal parameters inferred by inverse optimization are visualized in the following figures, which show that different weights are assigned to different edges. For example, the district "0" of higher value motivates the repositioning of idle vehicles towards this area, particularly along paths with low travel costs. Thus, a significant penalty is applied to insufficient rebalancing flows on the corresponding routes. The yellow line in the figure, which denotes high inferred penalty weights, achieves this motivation. In scenario 2, the situation becomes more complex, but we can still notice some "important" routes and the "not important" ones are not penalized.

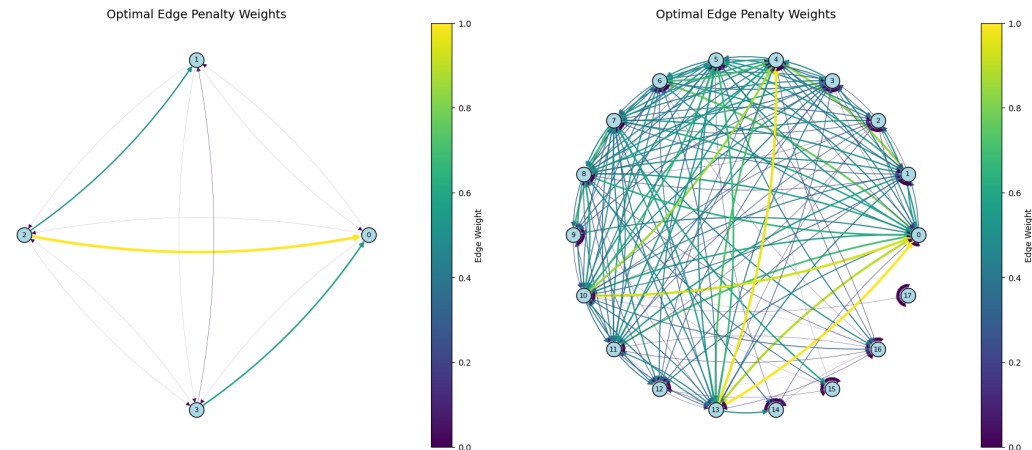

Figure 2: Optimal Weights in Scenario 1  Figure 3: Optimal Weights in Scenario 2

**Interpretations of Outputs.** To gain deeper insights into the effectiveness of the inverse method and explain model interpretability, we analyze how decisions made by two bi-level frameworks deviate from trajectories made by MPC. Table 6 shows that Bi-level-learned framework make decisions much closer to MPC trajectories, which indicates that the learned lower-level policy based on expert demonstrations extracts information efficiently compared with the original policy with short-sighted cost functions.

Table 6: Comparison of the cosine similarity and Manhattan distance between the rebalancing flow $f$ obtained from different bi-level models and the flow generated by the MPC.

| Formulation | Cosine Similarity | Manhattan Distance |
|---|---|---|
| Bi-level-unchanged | 0.477 | 44 |
| Bi-level-learned | **0.976** | **7** |

## C.4 SENSITIVITY ANALYSIS

In this subsection, we analyze how the number of ReLU terms included in the cost function and the noise in expert data influence the performance of inverse optimization. We use cosine similarity and Manhattan distance between the noise-free expert decisions and decisions made by solving the lower-level optimization problem with different settings.

- Sensitivity to Number of ReLU Terms

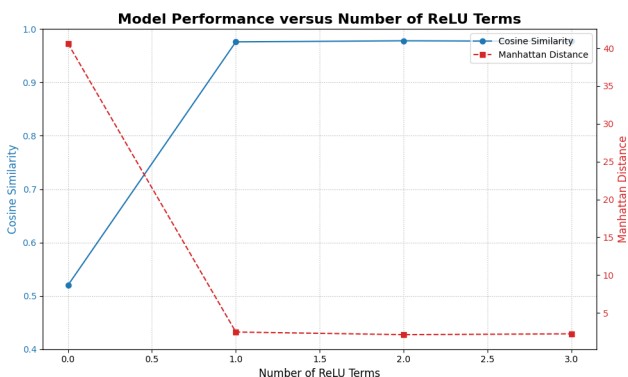

Figure 4: Sensitivity to the Number of ReLU Terms

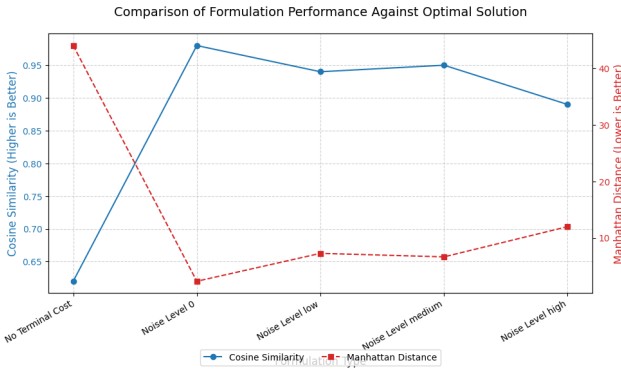

Figure 5: Sensitivity to Noise in Offline Data

Based on Figure 4, we notice that the quality of solutions doesn't vary a lot as number of ReLU terms varies. We can infer well-performed parameters via inverse optimization utilizing only a handful of expert decisions and very few ReLU terms.

- Sensitivity to Data Noise

Sometimes given demonstrations are not optimal, thus we figure out how the noise affect inverse optimization. As figure 5 shows, the solution is much closer to optimal expert demonstrations although some noise exists in offline data. Moreover, when we utilize noisy expert demonstrations to derive lower-level problem formulation and apply it in the bi-level framework, the framework improves the reward compared to the one achieved by original noisy expert demonstrations by $1.7\%$. Although the improvement is not so significant, it suggests a promising direction that we can employ our inverse optimization-guided bi-level framework to extract insights from sub-optimal offline data and make better decisions.

## C.5 SOLVE-TIME SCALING FOR INVERSE OPTIMIZATION

We systematically vary the problem dimension, the number of ReLU terms in the objective, and the number of expert demonstrations, and plot the resulting wall-clock solve times as shown in Figure 6.

We have observed that solve-time increases predictably with problem dimensions and the number of demonstrations, remaining tractable for typical system sizes without imposing prohibitive bottlenecks. The primary driver of complexity is the number of ReLU terms $(K)$. While $K = 1$ is computationally lightweight, increasing $K$ introduces significant combinatorial complexity. To address this, we reformulate the inverse problem as a Mixed-Integer Program (MIP) by treating the auxiliary dual variables associated with ReLU activation as binary. This reformulation is math-

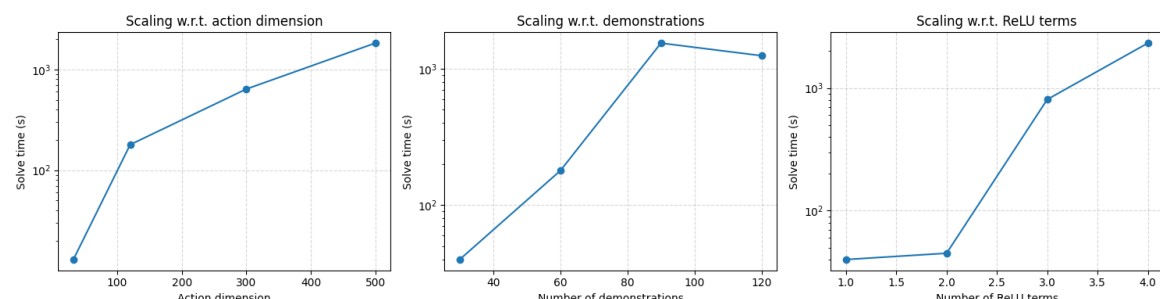

Figure 6: Solve-time Scaling Curve

ematically rigorous—aligning with the inherent discrete logic of ReLU functions—and leverages efficient branch-and-bound solvers to drastically accelerate convergence without compromising solution quality. We also note that solve times can vary even for instances of identical size, driven by the specific geometry of the expert demonstrations and the complexity of the active constraint set required to rationalize the data.

## D   USE OF LLMS

We utilized a large language model (LLM) to assist with improving the clarity, grammar, and overall readability of the text. The use of the LLM was strictly limited to language editing and refinement. All scientific contributions, including the core ideas, the methodological framework, the experimental design, and the mathematical derivations, are the original work of the authors.

