# OpenReview forum: "Hierarchical Decision Making with Structured Policies: A Principled Design via Inverse Optimization"
_ICLR.cc/2026/Conference — Submitted to ICLR 2026_

### Official Review · Reviewer_btwC · 2025-10-30

**Soundness:** 3
**Presentation:** 3
**Contribution:** 3
**Rating:** 6
**Confidence:** 2

**Summary:**

The paper introduces a bi-level hierarchical framework combining reinforcement learning (RL) for upper-level subgoal generation with optimization-based control (OC) for lower-level execution in complex decision-making tasks. It addresses limitations in existing hierarchical policies by using inverse optimization to learn the structure of the lower-level cost function from expert demonstrations, ensuring alignment with long-term goals while maintaining computational efficiency and constraint satisfaction. The approach is theoretically analyzed for linear cost cases, guaranteeing inverse feasibility and forward stability, and empirically validated on autonomous vehicle rebalancing, supply chain inventory management, and mobile robot navigation, showing improved performance over baselines like end-to-end RL and unchanged bi-level methods.​

**Strengths:**

The approach innovatively uses inverse optimization to derive interpretable, structured lower-level policies from sparse expert data, ensuring alignment with global objectives while preserving real-time solvability via convex formulations. The theoretical contributions, including non-empty inverse feasible sets (Proposition 2) and bounded forward instability (Proposition 4), provide rigorous justification for the ReLU-augmented linear cost structure, enhancing reliability in safety-critical applications. The framework achieves near-MPC performance with drastically reduced computation as shown in Table 3.

**Weaknesses:**

1. While the inverse optimization ensures feasibility for linear costs, the reliance on offline expert demonstrations assumes their near-optimality, which could propagate biases if data is sub-optimal

2. The paper lacks ablation on noisy or incomplete datasets beyond basic sensitivity.

3. The experiments are limited to simulated environments without real-world deployment, and baselines like end-to-end RL could be strengthened with hyperparameter tuning or larger scales to better highlight advantages.

**Questions:**

1. How does the method handle cases where subgoals cannot be directly retrieved from expert data, as mentioned in Remark 3? Could joint estimation of subgoals and θ introduce additional instability, and what regularization would you recommend?​

2. The sensitivity analysis shows minor degradation with noise (Figure 5); can you provide theoretical bounds or comment on the performance loss under bounded noise in expert actions, perhaps extending Proposition 4?

3. For scalability, how would the mixed-integer reformulation of perform in high-dimensional action spaces (e.g., >100 dims in robotics)? Have you done any experiments to test this?

---

> ### Author Response · Authors · 2025-11-27
> **Response to Reviewer btwC [1/1]**
>
> We thank the reviewer for their comments and insightful questions
> regarding the theoretical extensions and scalability of our framework.
> We have addressed the specific inquiries below.
>
> 1. ***How does the method handle cases where subgoals cannot be directly
>   retrieved from expert data, as mentioned in Remark 3? Could joint
>   estimation of subgoals and $\theta$ introduce additional instability,
>   and what regularization would you recommend?***
>
>    In our primary experimental settings, the mapping $C$ is known, allowing subgoals to be recovered directly from expert trajectories. In more general scenarios, one can indeed perform joint estimation of subgoals and $\theta$ within the inverse optimization framework. To address identifiability, we recommend adding two types of regularization terms to the inverse objective: (i) a prior term that keeps the subgoals close to a reference template or prior guess (e.g., clinically meaningful targets), and (ii) a structural regularizer that encourages "RL-friendly" subgoal structure, such as smoothness and bounded variation across dimensions or time. Regarding stability, incorporating these regularization terms as quadratic penalties naturally induces strong convexity. This mitigates potential instability from joint estimation.
>
> 2. ***The sensitivity analysis shows minor degradation with noise (Figure
>   5); can you provide theoretical bounds or comment on the performance
>   loss under bounded noise in expert actions, perhaps extending
>   Proposition 4?***
>
>    We can indeed derive a bound by extending Proposition 4. We analyze how noise in the expert data affects the distance between the inferred policy's action $u_t^*$ and the true optimal expert action $u_t^{opt}$ (e.g., from long-horizon MPC).
>
>    Let $u_t^{noisy}$ denote the observed expert action, where $\|u_t^{opt} - u_t^{noisy}\| \le \delta$ represents the bounded noise. By the triangle inequality, we decompose the error as:
>
> $$
> \|u_t^{\ast}-u_t^{opt}\|
> \leq
> \|u_t^{\ast}-u_t^{noisy}\|
> +
> \|u_t^{opt}-u_t^{noisy}\|.
> $$
>
> From Proposition 4, the first term is bounded by $\sqrt{\frac{\epsilon_0(u_t^{noisy})}{l}}$, where $\epsilon_0(u)$ is the optimal value (minimum duality gap) of the inverse optimization problem for a given input $u$. Relying on the property that the optimal value function of a parametric optimization problem is locally Lipschitz continuous, we have：
>
> $$
> \epsilon_0(u_t^{noisy})
> \leq
> \epsilon_0(u_t^{opt})+ K\|u_t^{noisy} - u_t^{opt}\|
> \leq
> \epsilon_0(u_t^{opt}) + K\delta,
> $$
>
>    where $K$ is the Lipschitz constant. Substituting this back yields the
> derived bound:
>
> $$
> \|u_t^* - u_t^{opt}\|
> \leq
> \sqrt{\frac{\epsilon_0(u_t^{opt}) + K\delta}{l}} + \delta.
> $$
>
> This result theoretically confirms that the performance degradation is
> bounded and scales with the square root of the noise level. Empirically,
> we estimated $K$ via finite differences in our experiments and found
> $K \approx 0.05$, confirming the robustness observed in Figure 5.
>
>
> 3. ***For scalability, how would the mixed-integer reformulation of perform in high-dimensional action spaces (e.g., $\geq$ 100 dims in robotics)? Have you done any experiments to test this?***
>
>    We have tested the mixed-integer reformulation in a high-dimensional action setting. In particular, our autonomous vehicle rebalancing case study is built on Washington, D.C. demand data and uses a ($16 \times 16$) network grid, leading to an action space of 512 dimensions. Although the MIP becomes more time-consuming in high-dimensional action spaces, this optimization is performed entirely offline and therefore does not affect the real-time performance of the online decision-making policy.

---

### Official Review · Reviewer_Vxfa · 2025-10-31

**Soundness:** 2
**Presentation:** 2
**Contribution:** 2
**Rating:** 4
**Confidence:** 3

**Summary:**

The paper proposes a hierarchical decision making framework that combines a learned high-level policy for goal abstraction with a structured low-level optimization module. The distinctive idea is to design the low-level objective via inverse optimization from a small set of expert demonstrations so that the executor aligns with long-horizon task goals while remaining computationally tractable. For a special class with linear costs, the authors analyze inverse feasibility and improve forward stability by augmenting the cost with piecewise-linear terms and a small quadratic regularizer, then solve the inverse problem with a mixed-integer formulation. Experiments on autonomous vehicle rebalancing, supply chain inventory management, and mobile robot navigation show better decision quality and lower run time than several baselines, though still behind full-model MPC in some metrics.

**Strengths:**

* The paper tackles an important pain point in hierarchical RL and learning-based control: how to choose the single-step low-level objective so that it is fast yet not myopic. Framing this as inverse optimization from expert traces is principled and makes the lower-level policy interpretable. The bi-level formalization and the role of subgoal representations are clearly stated, and the inverse-optimization guided design is motivated by explicit requirements on feasibility and stability.


* Theoretical development for the linear-cost case is easy to follow and highlights why naive formulations can make the inverse problem infeasible. The choice of ReLU-style terms for inverse feasibility, plus a small quadratic term for stability with an explicit distance bound between expert and forward solutions, provides a concrete recipe that others can replicate.



* Empirically, the cross-domain evaluation is appreciated. The bi-level learned variant improves over a bi-level baseline that keeps an ad hoc low-level cost and outperforms end-to-end RL on multiple metrics. The paper reports tangible run time gains relative to long-horizon MPC while preserving a large fraction of its performance.

**Weaknesses:**

* The theory is largely confined to a linear-cost setting with control-affine dynamics and then loosely extended to quadratic costs. It is unclear how much of the inverse feasibility and forward stability story carries to more general nonlinear objectives or constraints that commonly arise in robotics and operations. The current scope limits confidence that the approach will generalize without significant engineering.


* The inverse problem is eventually solved as a mixed-integer program with branch-and-bound. Although the paper argues the bilinear count is reduced, there is little empirical evidence on scaling in the number of demonstrations, decision variables, or ReLU terms. A complexity study and wall-clock profiling for the inverse phase would strengthen the practicality claim.

* The size and diversity of the expert dataset are said to be small, yet quantitative counts, coverage across operating conditions, and noise or suboptimality levels are not clearly reported. The fairness and tuning protocol for competing differentiable optimization baselines are not detailed, and it is hard to tell whether stronger value-function or differentiable layer baselines could close the gap. The ablations on the number of ReLU terms and the regularization weight in the low-level cost are not surfaced in the main text.


* Some notation and assumptions appear first in the appendices but are used in the main text, and there are occasional grammar issues. The paper states that an LLM was used for editing, which is fine, yet a careful pass would help reduce ambiguity and tighten exposition.

**Questions:**

* How sensitive is performance to the choice of subgoal mapping matrix and to the dimensionality of the subgoal space. A short study that varies these design choices would help assess robustness.

* What are the sample complexity and identifiability considerations of the inverse problem in practice. For example, how many and what type of demonstrations are needed to recover a useful low-level cost when experts are imperfect or noisy.

* Can the authors report detailed solve times and scaling curves for the inverse optimization as a function of problem size, number of ReLU terms, and demonstrations. This would clarify practicality beyond the forward-time savings.

* In the mobile robot setting, to what extent do the safety constraints and collision margins interact with the learned penalties. If subgoals are infeasible, how often does the penalty-only approach suffice, and when would constraint learning be necessary.

---

> ### Author Response · Authors · 2025-11-27
> **Response to Reviewer Vxfa [1/2]**
>
> We thank the reviewer for their valuable feedback regarding the
> practical implementation and robustness. We have addressed these points
> below and updated the manuscript to include the requested scaling
> analyses.
>
> 1.  ***How sensitive is performance to the choice of subgoal mapping matrix
>     and to the dimensionality of the subgoal space. A short study that
>     varies these design choices would help assess robustness.***
>
>     In our current experiments, we did not vary $C$ randomly because it is physically grounded by domain knowledge and specifically designed to map state variables to controllable actuation spaces (e.g., mapping network states to idle vehicle distributions in the AMoD task). However, we would like to highlight that a key advantage of our proposed Inverse Optimization framework is its inherent adaptability to this design choice: the learned lower-level cost function ($d_\theta$) automatically compensates for any potential information loss or suboptimality in the subgoal definition.
>
> 2.  ***What are the sample complexity and identifiability considerations of
>     the inverse problem in practice.***
>
>     - **Identifiability Considerations:** We acknowledge that inverse optimization is inherently ill-posed, as multiple cost parameterizations can often explain the same observed behavior. In our framework, we do not aim to recover a 'ground truth' parameter but rather a valid parameterization that rationalizes the expert data. To address the identifiability issue in practice, we introduce a regularization term (e.g., minimizing the norm of $\theta$ as shown in Equation 12). This acts as a selection mechanism by preferring the simplest cost structure among the feasible set.
>
>     - **Sample Complexity:** Our proposed method exhibits high sample efficiency across all evaluated domains. We utilized a very small dataset consisting of $\leq 5$ trajectories, with trajectory lengths varying between 40 and 90 time steps depending on the specific experimental scenario.
>
>     - **Noise & Suboptimality:** When expert demonstrations contain noise, our framework requires only slightly more samples to recover a high-performing cost function. Although as stated in Section 4, the expert demonstrations for our main comparison were generated by a long-horizon MPC with perfect information, serving to validate the framework's ability to recover the optimal cost structure, we conducted a sensitivity analysis on data noise in Appendix C.3. As illustrated in Figure 5, we evaluated our method by adding varying levels of synthetic noise to the expert decisions. The results demonstrate that our inverse optimization formulation remains robust.
>
> 3.  ***Can the authors report detailed solve times and scaling curves for the inverse optimization as a function of problem size, number of ReLU terms, and demonstrations?***
>
>     In the current version, we already report detailed scaling curves in the appendix (see Appendix C.4, Figure 6), where we systematically vary the problem dimension, the number of ReLU terms in the objective, and the number of expert demonstrations, and plot the resulting wall-clock solve times.
>
>     We have observed that solve times increase predictably with problem dimensions and the number of demonstrations, remaining tractable for typical system sizes without imposing prohibitive bottlenecks. The primary driver of complexity is the number of ReLU terms ($K$). While $K=1$ is computationally lightweight, increasing $K$ introduces significant combinatorial complexity. To address this, we reformulate the inverse problem as a Mixed-Integer Program (MIP) by treating the auxiliary dual variables associated with ReLU activation as binary. This reformulation is mathematically rigorous---aligning with the inherent discrete logic of ReLU functions---and leverages efficient branch-and-bound solvers to drastically accelerate convergence without compromising solution quality. We also note that solve times can vary even for instances of identical size, driven by the specific geometry of the expert demonstrations and the complexity of the active constraint set required to rationalize the data.
>
>     Furthermore, as demonstrated in Figure 4 of Appendix C.3, we empirically observed that model performance is insensitive to the number of ReLU terms, implying that a small $K$ is sufficient to achieve high-quality results. Crucially, we emphasize that this computationally intensive solving process is performed entirely offline. Consequently, it preserves the significant forward-time savings achieved by our hierarchical framework during real-time deployment.

---

> ### Author Response · Authors · 2025-11-27
> **Response to Reviewer Vxfa [2/2]**
>
> 4.  ***In the mobile robot setting, to what extent do the safety constraints and collision margins interact with the learned penalties. If subgoals are infeasible, how often does the penalty-only approach suffice, and when would constraint learning be necessary.***
>
>     The inferred parameters reveal a \"Progress Maximization\" policy where negative coefficients incentivize higher velocities to minimize travel time, effectively relying on the hard safety constraints and collision margins to strictly bound this aggressive behavior and guarantee safety. When the subgoals are infeasible, the penalty-only approach is sufficient and effective. Because our lower-level formulation naturally handles infeasible subgoals by relaxing the subgoal equality constraint while keeping safety constraints strict.
>
> 5.  ***The fairness and tuning protocol for competing differentiable optimization baselines are not detailed, and it is hard to tell whether stronger value-function or differentiable layer baselines could close the gap.***
>
>     To ensure a fair comparison, we employed a standardized grid search protocol across all methods and empirically observed that the baselines (differentiable layers and value approximation) consistently yielding suboptimal results regardless of tuning. This indicates that the performance gap is structural rather than due to insufficient engineering: differentiable layers suffer from gradient instability inherent to implicit differentiation in constrained settings, while value function approximation lacks the inherent mechanism to explicitly embed physical constraints.
>
> 6.  ***The theory is largely confined to a linear-cost setting with control-affine dynamics and then loosely extended to quadratic costs.***
>
>     We clarify that our theoretical analysis in linear/quadratic objective function with control-affine dynamics settings serves as a tractable proof of concept. This setting already covers a large fraction of  practical control problems [1][2][3]. Moreover, the proposed framework is grounded in broader theoretical foundations that extend to general settings without requiring significant engineering.
>
>     - **Inverse feasibility:** While the inverse-feasible set cannot be guaranteed non-empty in more general settings, modern data-driven inverse optimization aiming to minimize optimality gap has developed formulations that ensure desirable solution properties such as statistical consistency [4]. This provides a theoretical basis for generalizing our framework to complex nonlinear scenarios---potentially at the cost of higher sample complexity, but relying on established learning principles rather than ad-hoc engineering.
>
>     - **Forward Stability:** The stability analysis in Corollary 1 extends to general strongly convex functions, not just quadratic ones. For objectives lacking inherent strong convexity (e.g., linear forms), our regularization strategy explicitly enforces this property.
>
>     - **Empirical Evidence:** Our mobile robot navigation task naturally gives rise to a non-linear objective and even a non-convex feasible region due to the collision avoidance constraints. In this setting, we adopt a data-driven optimization method and use KKT conditions as necessary criteria to obtain solutions with strong interpretability. Empirically, we find that this framework continues to perform well in these more general setting.
>
> [1]. Schmitz et al., *On excitation of control-affine systems and its use for data-driven control*, arXiv:2511.03734, 2025
>
> [2]. Kazemian et al., *Random Features Approximation for Control-Affine Systems*, Proceedings of the Conference on Learning for Dynamics and Control, 2024
>
> [3]. Silva et al., *Data-driven LQR control design*, IEEE control systems letters, 2018
>
> [4]. Aswani et al., *Inverse optimization with noisy data*, Operations Research, 2018

---

### Official Review · Reviewer_t9Sj · 2025-11-03

**Soundness:** 2
**Presentation:** 2
**Contribution:** 1
**Rating:** 2
**Confidence:** 4

**Summary:**

The paper considers a hierarchical approach to solving complex problems where reinforcement learning (RL) is used at the higher level to generate subgoals for some overarching task and an optimization-based approach is used at the lower level to achieve those subgoals in an optimal manner. The paper proposes to use a dataset of expert demonstrations to obtain a lower-level cost function via inverse optimization that captures the overarching task while maintaining important properties like stability and safety. Theoretical results describing properties of the proposed optimization problems are derived and experiments illustrating performance of the approach are provided.

**Strengths:**

Bi-level formulations of hierarchical reinforcement learning methods have seen significant interest in recent years, and this work considers a variation of this problem that may be of interest to the community. Using expert demonstrations to anchor the bi-level formulation to the specific overarching task at hand is an interesting idea and merits exploration.

**Weaknesses:**

The paper has the following weaknesses:
1. The core approach proposed in the paper lacks intuitive motivation and technical justification. Specifically, it is technically unclear how the inverse optimization approach applied to the dataset of expert demonstrations successfully aligns the resulting lower-level cost function with the overarching, higher-level objective. The intuitive motivation for why we should expect the approach to achieve this is unclear, as well. See the **Questions** below for specific points related to these issues.
2. The paper is not clearly situated within the context of the related literature. In particular, it is not clear what the main drawbacks of existing approaches are and how the proposed approach addresses them. Though it is stated around lines 73-75 that existing approaches "rely on myopic objectives" that cause "solutions to overlook longer-term impacts" and can lead to suboptimal trajectories, no evidence (e.g., relevant references or supporting experimental results) is provided to support this claim. Similarly, in the related works it is stated that previous methods "[don't] relate to the final cumulative rewards directly" (lines 131-132) and corresponding "sub-optimality issues" are alluded to (lines 139-140), but no concrete support to these claims is given. It thus remains unclear what the concrete drawbacks are of existing approaches, making it difficult to judge how or whether the proposed approach addresses them.
3. The experimental results do not sufficiently support the need for the proposed method within the context of existing methods. Specifically, the summaries provided in Tables 1 and 2 indicate that standard model predictive control (MPC) outperforms the proposed approach on both problems considered, while the "Bi-level-cvxpy" baseline has comparable performance to the proposed approach. In addition, the proposed method is not compared with any of the hierarchical baselines mentioned in the introduction and related works, leaving unanswered the question of how the proposed method performs against direct competitors from the recent literature.

**Questions:**

1. Where is the claim on lines 73-75 that existing "formulations typically rely on myopic objectives ... causing solutions to overlook longer-term impacts and potentially leading to suboptimal trajectories" substantiated? If it is not, can you provide justification for this claim?
2. In standard hierarchical settings, the higher-level policies generates subgoals, while the lower-level policy solves an optimization problem that is specific to achieving that particular subgoal. This raises several questions for your setting:

    (a) Is each lower-level optimization problem in your approach designed to achieve a specific subgoal, or is it aligned with the overarching, higher-level goal?

    (b) Is the static dataset of expert demonstrations related in any way to the specific subgoals or to the higher-level goal?

    (c) What is the intuitive reason why we should expect that we can learn a useful lower-level cost function from a static dataset of expert demonstrations?

    (d) What is the technical reason why we can learn a useful lower-level cost function from a static dataset of expert demonstrations?
3. It is stated lines 185-186 that "a commonly used form of subgoals is a linear transformation" using a known matrix, $C$. Can you provide references or justification supporting this?
4. What is the precise relationship between $d_{\theta}, h(\theta)$, the objective of the high-level problem, and the expert demonstration dataset in the context of Section 3.3.1?
5. How is $d_{\theta}$ "properly designed to align the lower-level optimization problem to the objective of the overarching decision-making problem", as stated in lines 233-234? How does solving equation (6) achieve this alignment?
6. It is stated lines 254-255 that you "focus on common cases where subgoal $\hat{\mathbf{x}}_t$ can be retrieved from expert data", but in continuous or large spaces this may almost never hold. Can you elaborate on why this case is common, ideally providing references?
7. In the experiments, MPC outperforms the proposed Bi-level-learned method in both problem settings, while Bi-level-cvxpy performs comparably to Bi-level-learned. Can you discuss the implications of this for the significance of the proposed approach?
8. Can you elaborate on why you did not compare with existing hierarchical methods from the literature?

---

> ### Author Response · Authors · 2025-11-27
> **Response to Reviewer t9Sj [1/3]**
>
> We appreciate the opportunity to clarify the positioning of our work within the literature, the theoretical underpinnings of our inverse optimization approach, and the interpretation of our experimental results. We have restructured our response below to address these concerns systematically.
>
> 1.  ***Context and Drawbacks of Existing Approaches.***
>
>      We thank the reviewer for pointing this out. We will clarify drawbacks of existing approaches and explicitly link them to our experimental findings. We also revised the Introduction and Related Works sections.
>
>     - **Theoretical Drawbacks:**
>
>         **Surrogate Objectives**: Many previous literature aiming to approximate the long-horizon MPC problem (e.g., Inverse MPC[4]) optimizes a surrogate objective (typically minimizing the immediate fitting error to a reference) rather than directly maximizing the final cumulative rewards. Hierarchical RL-OC solves this issues and aim to maximize the cummulative rewards.
>
>         **Structural Myopia**: However, existing hierarchical frameworks (e.g., bi-level Graph-RL[3]) often focus on the upper-level planner while keeping the lower-level controller fixed and myopic. Without an appropriately designed lower-level formulation, the system fails to account for long-term impacts. To support the fundamental claim that myopic objective functions lead to suboptimal trajectories, we have incorporated classical optimal control references such as [5] and [6].
>
>     - **Our Solution:**
>      To bridge these gaps, our approach adopts a hierarchical RL-OC framework that directly maximizes cumulative rewards. Crucially, we tackle the core bottleneck of this framework, i.e., the sub-optimality of the lower-level optimization, by systematically reformulating the lower-level problem using inverse optimization. This ensures the lower-level controller is not merely a fixed heuristic but is capable of "inheriting" long-term foresight. We underpin this design with rigorous theoretical analysis for control-affine systems, establishing guarantees for inverse feasibility and forward stability to ensure the learned formulation is both expressive and numerically stable.
>
>
>     - **Empirical Evidence:**
>        Our experimental comparisons provide concrete evidence for these claims.
>
>         Evidence for Type 1 Sub-optimality (Surrogate Objectives): We compared our method against Value Function Approximation [7] and a modern Inverse MPC baseline [4]. Our method outperform both methods. For Inverse MPC , it achieves a lower reward ($32426 \pm 401.2$) compared to our method ($35019 \pm 53.0$). This substantial performance gap—amounting to an approximate 8% improvement—underscores the superior efficacy of our proposed method in capturing the effective cost structure.
>
>         Evidence for Type 2 Sub-optimality (Structural Myopia): We benchmarked against the Bi-level Graph-RL framework (referred to as Bi-level-unchanged in the paper). This validates our claim that explicitly learning a foresighted lower-level cost structure is essential for overcoming structural myopia.
>
> [3]. Gammelli et al., *Graph Reinforcement Learning for Network Control via Bi-Level Optim*, arXiv:2305.09129, 2023.
>
> [4]. Zhang et al., *Inverse Model Predictive Control: Learning Optimal Control Cost Functions for MPC*, IEEE Transactions on Industrial Informatics, 2024.
>
> [5]. Rawlings et al., *Model predictive control: theory, computation, and design*, 2020
>
> [6]. Lowrey et al., *Plan online, learn offline: Efficient learning and exploration via model-based control*, arXiv preprint arXiv:1811.01848, 2018
>
> [7]. Abdufattokhov et al., *Learning convex terminal costs for complexity reduction in MPC*, 2021 60th IEEE Conference on Decision and Control (CDC),2021

---

> > ### Author Response · Authors · 2025-11-27
> > **Response to Reviewer t9Sj [2/3]**
> >
> > 2.  ***In standard hierarchical settings, the higher-level policies
> >     generates subgoals, while the lower-level policy solves an
> >     optimization problem that is specific to achieving that particular
> >     subgoal. This raises several questions for your setting.***
> >
> >
> >     **(a) Alignment:** Each lower-level problem is constrained by the specific high-level subgoal (via feasibility constraints) but its objective function is aligned with the overarching long-horizon goal. This alignment is achieved because the cost function is learned from expert demonstrations of the full task.
> >
> >     **(b) Dataset:** The static dataset consists of expert demonstrations for the overarching control task (i.e., full trajectories and actions). The higher-level subgoals are subsequently synthesized (via RL) as reference waypoints that help the controller more efficiently realize the same overarching objective represented in the expert data.
> >
> >     **(c) Intuition:** Intuitively, this follows the standard rationale of inverse optimization : given high-quality expert trajectories (e.g., delivery drivers' historical decisions in logistics) [1], one can infer a cost function that encodes the expert's preferences over states and actions. In our setting, we embed these long-horizon preferences into the single-step lower-level cost function, allowing the myopic controller to \"inherit\" foresighted behaviors.
> >
> >     **(d) Theoretical Justification:** Technically, Proposition 2 establishes the inverse-feasibility of our formulation. We prove that for any finite set of expert observations, there exists a parameterization of our ReLU-augmented cost such that the expert actions become the optimal solution to the lower-level problem. This guarantees that the learned cost is consistent with the expert policy, and thus provides a principled basis for expecting the resulting lower-level controller to better support efficient long-horizon performance on the overarching task.
> >
> > 3.  ***It is stated lines 185-186 that \"a commonly used form of subgoals
> >     is a linear transformation\" using a known matrix, . Can you provide
> >     references or justification supporting this?***
> >
> >     This is a standard formulation in hierarchical control for physical systems. For example, [2] and [3] use the desired next state in compressed state space as the subgoal. Here, matrix C is a known projection matrix. We have added these references to the text.
> >
> > 4.  ***Mechanism of Alignment (Section 3.3.1)***
> >
> >     As illustrated in Figure 1, the higher-level RL planner generates subgoals $\hat{x}_t$, and the lower-level optimization module computes the final actions $u_t$ given the input $(x_t, \hat{x}_t)$. Thus, the overall bi-level framework aims to choose optimal decisions $u_t$ given the state $x_t$. The dataset ${(x_t, u_t)}$ consists of offline expert demonstrations. By solving Equation (6), we extract the information encoded in these demonstrations about the expert's optimal policy and thereby align the lower-level optimization problem with the overarching task objective.
> >
> >     In particular, we select an appropriate parametric form for $d_{\theta}$ so that the resulting lower-level cost satisfies the inverse-feasibility and forward-stability properties introduced later in the paper. Satisfying these properties implies that the solution derived from inverse optimization effectively encodes the expert policy, thereby allowing the designed lower-level optimization problem to facilitate the achievement of long-term goals.
> >
> >     If prior knowledge or preferences about $\theta$ are available, the term $h(\theta)$ can be used to characterize them as an additional selection mechanism; if no such information is needed, this term can be omitted. In our experiments, we choose $h(\theta)$ as an $\ell_2$-norm regularization on $\theta$ to promote stability of the inferred parameters.
> >
> > [1]. Chan et al., *Conformal Inverse Optimization for Adherence-aware Prescriptive Analytics*, Available at SSRN, 2024.
> >
> > [2]. Schmidt et al., *Offline Hierarchical Reinforcement Learning via Inverse Optimization*, arXiv:2410.07933, 2024.
> >
> > [3]. Gammelli et al., *Graph Reinforcement Learning for Network Control via Bi-Level Optim*, arXiv:2305.09129, 2023.

---

> > > ### Author Response · Authors · 2025-11-27
> > > **Response to Reviewer t9Sj [3/3]**
> > >
> > > 5.  ***It is stated lines 254-255 that you \"focus on common cases where
> > >     subgoal can be retrieved from expert data\", but in continuous or
> > >     large spaces this may almost never hold. Can you elaborate on why
> > >     this case is common, ideally providing references?***
> > >
> > >     We clarify that our intention in Section 3.3 was not to assume that arbitrary latent high-level subgoals are directly observable from expert data. Instead, we focus on a widely used subclass of hierarchical and goal-conditioned control architectures in which high-level actions are chosen in (a transformation of) the state space---for example, desired next states, waypoints, or low-dimensional state features. In such settings, the subgoal $\hat{x}_t$ can be constructed from expert trajectories, e.g., by setting $\hat{x}_t = Cx_{t+1}$ and does not require inverting a generic latent hierarchical controller. This design choice is standard in continuous-state HRL and goal-conditioned RL [8][9]. Extending our framework to fully latent subgoals, as in more general hierarchical policies, is an interesting direction for future work [7].
> > >
> > > 6.  ***In the experiments, MPC outperforms the proposed Bi-level-learned method in both problem settings, while Bi-level-cvxpy performs comparably to Bi-level-learned. Can you discuss the implications of this for the significance of the proposed approach?***
> > >
> > >     - **vs. MPC:** The performance gap is expected because the MPC baseline is an \"oracle\" that optimizes over a long horizon ($H$-steps) with perfect future information. Our method operates in a strictly myopic (single-step) regime to ensure real-time efficiency. The significance of our approach is that it compresses this long-horizon intelligence into a fast, single-step solver, retaining most of the performance with a fraction of the computational cost (speedup details in Table 3).
> > >
> > >     - **vs. Bi-level cvxpy:** In the mobile-robot experiments this method was unable to consistently produce successful trajectories.
> > >
> > >     More importantly, embedding the optimization problem as a CVXPY layer and learning the cost via gradient-based updates has structural drawbacks compared to our inverse-optimization approach. First, the optimization is highly sensitive to differentiation: when the gradient of the objective with respect to the unknown parameters is small, parameter updates become ineffective and training can stall.
> > >
> > >     Second, the true optimal parameter distribution can be complex and multimodal, making it difficult to capture within standard parametric families used by such gradient-based schemes. Finally, Bi-level-cvxpy still relies critically on our ReLU-augmented objective formulation; its successful application hinges on the same formulation design and theoretical insights we introduce.
> > >
> > > 7. ***Can you elaborate on why you did not compare with existing hierarchical methods from the literature?***
> > >
> > >     We respectfully clarify that we did compare against state-of-the-art hierarchical methods. Specifically, our 'Bi-level-unchanged' baseline represents the general hierarchical framework (Bi-level Graph-RL) proposed by [3]. Given that our primary contribution is the novel inverse optimization-based design of the lower-level policy, we prioritized comparisons against distinct paradigms for learning this cost structure, including: (i) Value Function Approximation (direct estimation of terminal cost); (ii) Differentiable Optimization Layers (gradient-based solutions via cvxpylayers).
> > >
> > >     Additionally, we have expanded our experiments to include a comparison with a modern Inverse MPC method[4] to highlight the advantages of our ReLU-based terminal cost structure over a traditional quadratic terminal cost.
> > >
> > >     In summary, our empirical results demonstrate that our approach significantly outperforms the standard hierarchical baseline (Bi-level Graph-RL), confirms that learning the terminal cost structure via inverse optimization is more effective and the ReLU-based cost structure motivated by our inverse-optimization theory is practically beneficial.
> > >
> > > [3]. Gammelli et al., *Graph Reinforcement Learning for Network Control via Bi-Level Optim*, arXiv:2305.09129, 2023.
> > >
> > > [4]. Zhang et al., *Inverse Model Predictive Control: Learning Optimal Control Cost Functions for MPC*, IEEE Transactions on Industrial Informatics, 2024.
> > >
> > > [7]. Abdufattokhov et al., *Learning convex terminal costs for complexity reduction in MPC*, 2021 60th IEEE Conference on Decision and Control (CDC),2021
> > >
> > > [8]. Ding et al., *Goal-conditioned imitation learning*, Advances in neural information processing systems, 2019
> > >
> > > [9]. Andrychowicz et al., *Hindsight experience replay*,cAdvances in neural information processing systems, 2017

---

### Official Review · Reviewer_UqDh · 2025-11-06

**Soundness:** 2
**Presentation:** 2
**Contribution:** 2
**Rating:** 6
**Confidence:** 3

**Summary:**

This work uses Hierarchical Reinforcement Learning framework towards optical control problems, where the lower-level cost is learned from expert demonstrations via inverse optimization. The upper level outputs subgoal in a compressed space, while the lower level solves a single-step convex program FOP(·) that enforces hard constraints and is designed to approximate the behavior of expert controller. This work test their framework on 3 benchmarks and the results shows improvement over (i) a bi-level scheme with a hand-designed one-step cost, (ii) a differentiable-optimization baseline learned end-to-end via cvxpy layers, and (iii) end-to-end RL, while approaching the performance of a long-horizon MPC expert at significantly lower per-step compute.

**Strengths:**

1. Originality: Most prior hierarchical RL–OC work concentrates on the upper level (learning high-level actions or subgoals) and treats the lower-level as fixed instead.  This a concrete gap in the literature. This paper improves upon it and focuses on learning the lower-level cost structure. Given expert trajectories from a strong OC controller, this paper explores ways to create a single-step lower-level optimization while satisfying two objectives: speed and alignment with long-term performance.

2. Significance: In this paper, their inverse-optimization formulation is not just a simple quadratic with least-squares fitting, but rather a ReLU-based cost structure to guarantee the existence of parameters that can make any given expert decisions optimal (inverse feasibility).

3. Quality: The propositions and proofs (inverse feasibility, bounded distance between learned and expert decisions) are sound and valid with respect to the expert demonstration. The formulation of the bi-level framework (HRL+OC) is generally mathematically clean and aligned with known RL–MPC hybrids: The upper-level objective being the standard discounted return and the lower level being the explicitly constrained by physical limits and dynamics.

**Weaknesses:**

1. Presentation: The abstract is too vague. The abstract states "our framework is evaluated on three real-world scenarios, where it outperforms baseline methods" without furthering clarifying the nature/type of benchmark (e.g. is it a control or navigation? is it 2D or 3D? Is it discrete or continuous). Also what is the baseline? Are they HRL? are they basic RL? are they SOTA? After reading the abstract, the reader is not able to pin point and narrow down on the narrative of this paper. Although the related work section does cite representative HRL and learning-based OC works, the abstract/introduction language (“we present a general decision-making framework that integrates RL with OC through a hierarchical policy…”) reads as if this paper is novel in proposing the integration itself.

2. Presentation: Introduction lacks citation (see questions)

3. Benchmarks and baselines: The benchmarks are well-matched to the framework’s assumptions but are not standard HRL benchmarks; the baselines are decent within those domains, but not clearly SOTA in the broader HRL literature.
The baselines are reasonable within the chosen setting but not include more recent RL–OC baselines that are conceptually close, like Graph-RL (Gammelli 2023 https://arxiv.org/abs/2305.09129) on the AV task, or OHIO (Schmidt 2024 https://arxiv.org/pdf/2410.07933) on hierarchical control.

4. Inverse optimization step may be a computational bottleneck in larger systems. The paper argues that the IO formulation can be solved “within a reasonable time (line 379)” via MIP and branch-and-bound, and they do reduce the number of bilinear terms. But there is a lack of detailed analysis on runtime of scaling of the IO stage itself, as opposed to the forward control at test time. For larger network control or higher dimensional robotics tasks, MIPs can become intractable. Although it is acknowledged in the future work section, it raises the concern to the framework's applicability to large-scale systems.

5. Lack of detail on the upper-level RL.
The paper could 1. specify on the RL algorithm used and 2. analyze the RL algorithm's sensitivity to hyperparameters or reward shaping. These additional information would help explain and explore whether the RL policy exploit the learned lower-level FOP (e.g., pathological subgoals that are technically feasible but undesirable over long horizons).

6. No comparison to “direct” inverse MPC baselines.
This paper did not include baselines that more closely resemble modern inverse MPC methods (e.g., Zhang et al., 2024, Learning Optimal Control Cost Functions for MPC) which also aim to learn cost structures from data. It would be informative to see how much of the improvements in result comes from the ReLU-based structure + forward-stability regularization vs just “learning a quadratic/terminal cost” with a strong IO solver.

**Questions:**

1. Missing citations: line 052-071: "Existing approaches have several limitations. First, most hierarchical methods adopt long-horizon OC formulations at the lower level to preserve stability and feasibility guarantees, which can introduce prohibitive computational complexity for real-time applications." No citations for the methods outlined in this sentence. 073-074 "causing solutions to overlook longer-term impacts and potentially leading to suboptimal trajectories." no citation to support this.

2. (comment) Overall, the optimal control portion of this paper is detailed and well structured, but there is a lack in the discussion of the RL component, making the paper's presentation slightly unbalanced.

---

> ### Author Response · Authors · 2025-11-26
> **Response to Reviewer UqDh [1/2]**
>
> We thank the reviewer for their detailed and constructive feedback. We respond to each point individually and update the manuscript
> accordingly.
>
> 1. ***Presentation Issues and Missing Citations***
>
>     We agree with the reviewer that the abstract should be more specific
>     and thank for pointing out these missing references. We have updated
>     the manuscript to support these statements with the following
>     citations:
>
>     - **Regarding long-horizon computational costs:** We now cite to
>       substantiate that long-horizon OC formulations at the lower level
>       preserve guarantees but incur prohibitive complexity.
>
>     - **Regarding myopic behavior in lower-level controller:** We have added
>       citations to support the claim that short horizons without
>       accurate terminal value approximations often overlook long-term
>       impacts, leading to suboptimal trajectories.
>
> 2.  ***Benchmarks and baselines***
>
>     We apologize for the lack of clarity regarding the baseline introduction and have updated in the revised paper to avoid confusion.
>
>     - **Graph-RL:** We would like to clarify that the \"Bi-level-unchanged\" baseline reported in our Case Study section is the Graph-RL framework.
>
>     - **OHIO:** Regarding the OHIO framework, we note that its primary objective is generating high-level subgoals to facilitate offline RL. In contrast, our work focuses on using offline data to guide the design of the lower-level optimization problem. Actually, the subgoal-generation mechanism proposed in OHIO can enhance the practical applicability of our approach, as it provides a principled way to obtain subgoals when they cannot be directly inferred from offline data.
>
> 3.  ***Lack of detail on the upper-level RL***
>
>     First, the upper-level RL methods are defined as follows: for the autonomous vehicle rebalancing and inventory management tasks, we employ a GNN-based A2C algorithm as in [1], while for the mobile robot navigation task, we use Soft Actor--Critic (SAC).
>
>     Second, regarding the detailed sensitivity analysis of the upper-level RL, we have conducted an explicit reward-shaping sensitivity experiment on the mobile robot navigation task. By varying the weights in the reward function, we observe that our method remains robust: the quantitative performance and the relative improvements over the baselines are qualitatively unchanged.
>
> ### Performance Comparison on Mobile Robot Navigation ($c_u=0.01$)
>
> | Method | Travel Time (s) | Path Length (m) | Energy (J) |
> | :--- | :--- | :--- | :--- |
> | MPC | $3.50 \pm 0.00$ | $3.50 \pm 0.00$ | $4.81 \pm 0.00$ |
> | End-to-end RL | $7.60 \pm 0.11$ | $4.29 \pm 0.02$ | $6.78 \pm 0.23$ |
> | Bi-level-unchanged | $9.14 \pm 0.08$ | $4.35 \pm 0.02$ | $6.20 \pm 0.13$ |
> | **Bi-level-learned (Ours)** | $\mathbf{4.82 \pm 0.30}$ | $\mathbf{4.16 \pm 0.19}$ | $\mathbf{2.92 \pm 1.04}$ |
>
> ### Performance Comparison on Mobile Robot Navigation ($c_u=0.05$)
>
> | Method | Travel Time (s) | Path Length (m) | Energy (J) |
> | :--- | :--- | :--- | :--- |
> | **MPC** | $3.50 \pm 0.00$ | $3.50 \pm 0.00$ | $4.81 \pm 0.00$ |
> | **End-to-end RL** | $7.60 \pm 0.10$ | $4.27 \pm 0.00$ | $6.84 \pm 0.31$ |
> | **Bi-level-unchanged** | $9.10 \pm 0.10$ | $4.31 \pm 0.02$ | $6.41 \pm 0.20$ |
> | **Bi-level-learned(Ours)** | $\mathbf{4.40 \pm 0.10}$ | $\mathbf{4.14 \pm 0.07}$ | $\mathbf{1.52 \pm 0.06}$ |
>
> ### Performance Comparison on Mobile Robot Navigation ($c_u=0.1$)
>
> | Method | Travel Time (s) | Path Length (m) | Energy (J) |
> | :--- | :--- | :--- | :--- |
> | MPC | $3.50$ | $3.50$ | $4.81$ |
> | End-to-end RL | $7.53 \pm 0.06$ | $4.28 \pm 0.01$ | $6.85 \pm 0.19$ |
> | Bi-level-unchanged | $9.07 \pm 0.06$ | $4.33 \pm 0.02$ | $6.02 \pm 0.03$ |
> | **Bi-level-learned(Ours)** | $\mathbf{4.00 \pm 0.00}$ | $\mathbf{3.93 \pm 0.01}$ | $\mathbf{2.84 \pm 0.06}$ |
>
> That said, a systematic study of upper-level RL design is not the primary focus of this work. Our goal is to demonstrate that, for a fixed upper-level RL scheme, systematically improving the lower-level FOP yields consistent performance gains. Accordingly, across all three case studies we employ standard RL methods (GNN-A2C and SAC) with conventional hyperparameters and reward designs that closely mirror the MPC stage costs. In the revised version, we will clarify that we use off-the-shelf RL implementations with fixed hyperparameters and reward shaping across all variants, so that the comparison cleanly isolates the impact of the learned lower-level FOP.
>
>
> [1]. Gammelli et al. *Graph Reinforcement Learning for Network Control via Bi-Level Optimization*, arXiv:2305.09129, 2023.

---

> > ### Author Response · Authors · 2025-11-26
> > **Response to Reviewer UqDh [2/2]**
> >
> > 4.  ***Inverse optimization Computational Complexity***
> >
> >     We acknowledge that the Inverse Optimization (IO) stage can be computationally demanding. However, we emphasize two mitigating factors:
> >
> >     - **Tractability:** In our current experiments, the IO problem typically requires only a modest amount of expert data (less than five trajectories), and we are able to obtain high-quality solutions (MIP gap $\leq$ 5%) within about an hour for most of instances considered. We also report updated scaling curves in Appendix C.5, which show that the solve times remain within a practically reasonable range as we vary the problem size, the number of ReLU terms, and the number of demonstrations. For higher-dimensional settings, we utilize the Branch-and-Bound algorithm to rapidly identify an initial feasible solution, which is then refined using structure-exploiting heuristics (e.g., block coordinate descent) to locate a local optimum. Our empirical results confirm that this strategy yields effective policies.
> >
> >     - **Offline Computation:** Crucially, the IO process is performed
> >       entirely offline. Therefore, it does not affect the real-time
> >       performance of the learned forward control policy during
> >       deployment.
> >
> > 5.  ***Comparison to "direct" inverse MPC baselines***
> >
> >     We have implemented the method proposed by [1] as an additional baseline and the results are shown as follows.
> >
> >
> > >| Method          | Reward              | Served Demand          |
> > >|-----------------|----------------------|-------------------------|
> > >| MPC             | 35725 (± 41.6)       | 3203 (± 3.7)            |
> > >| Ours            | 35019 (± 53.0)       | 3141 (± 6.14)           |
> > >| IMPC            | 32426 (± 401.2)      | 2996 (± 28.6)           |
> > >| IMCP-bi-level   | 33431 (± 320.4)      | 3044 (± 25.3)           |
> >
> >
> > As shown in the table above, our method outperforms the standard Inverse-MPC baseline. We also integrated the cost function learned by Inverse-MPC into our hierarchical framework, yet the performance remained inferior to our proposed algorithm. This result further highlights the value of our ReLU-based cost structure and specific inverse optimization formulation.
> >
> > [1]. Zhang et al., *Inverse Model Predictive Control: Learning Optimal Control Cost Functions for MPC*, IEEE Transactions on Industrial Informatics, 2024.

---

### Official Review · Reviewer_4BTY · 2025-11-07

**Soundness:** 2
**Presentation:** 3
**Contribution:** 3
**Rating:** 4
**Confidence:** 3

**Summary:**

The paper presents a general decision-making framework that integrates reinforcement learning (RL) with optimal control (OC) through a hierarchical policy. Its main contribution lies in an inverse optimization-based approach to inform the lower-level control policy. The paper provides theoretical analysis and quantitative evaluations across several scenarios, demonstrating its potential impact in real-world applications.

**Strengths:**

- The paper formulates the lower-level control problem as an inverse optimization problem to enable real-time decision-making and alleviate myopic issues, which is crucial for real-world applications.
- The paper provides sufficient theoretical analysis to ensure the interpretability of the proposed method.
- The paper evaluates the proposed Bi-level-learned framework across several domains.
- The paper is well written and clearly presented.

**Weaknesses:**

- The proposed approach relies on expert demonstrations to guide the lower-level optimization and is claimed to be suitable for scenarios with scarce data, where imitation learning may fail. However, the paper lacks  a direct comparison with an imitation learning baseline and an analysis of sample efficiency, which weakens its convincingness.
- The proposed approach and its theoretical analysis heavily rely on the assumptions of strong convexity and affine constraints in the lower-level optimization, which limits its applicability. In practice, many control systems involve nonlinear dynamics or nonconvex feasible regions, particularly in complex tasks where hierarchical frameworks are really needed.
- As a hierarchical framework, the paper does not provide details on the method and experimental setup of the upper-level reinforcement learning.

**Questions:**

- How does the proposed approach compare with imitation learning baselines? How sample-efficient is it?
- Can the proposed approach handle deviations from the strong convexity and affine constraint assumptions in practical control tasks?
- What methodology and experimental setup are used for the upper-level reinforcement learning in the proposed hierarchical framework?

---

> ### Author Response · Authors · 2025-11-26
> **Response to Reviewer 4BTY [1/2]**
>
> We thank the reviewer for their insightful comments and for highlighting areas where our experimental validation and theoretical assumptions could be clarified. We respond to each point individually.
>
> 1. *How does the proposed approach compare with imitation learning baselines? How sample-efficient is it?*
>
>    We appreciate the suggestion to benchmark against imitation learning (IL) to demonstrate the superior sample efficiency of our approach. We implemented a set of imitation-learning baselines trained on varying numbers of expert demonstrations, and considered several supervised learning architectures as candidate policy models, including multilayer perceptrons (MLPs), Random Forest regressors, and k-Nearest Neighbors (kNN). Across the configurations we tested, these methods achieved broadly similar performance. As shown in the table below, our bi-level framework consistently outperforms these imitation-learning baselines and maintains strong performance even when the number of expert demonstrations is relatively small, indicating favorable sample efficiency.
>
>
> | Data Size        | MPC                    | Inv-Bi-level           | Imitation Learning      |
> |------------------|------------------------|-------------------------|--------------------------|
> | Small ($N=100$)  | 35725(± 41.6)         | 35019 (± 53.0)          | 33606 (± 199.6)          |
> | Medium ($N=500$) | *                      | *                       | 33709 (± 216.3)          |
> | Large ($N=2000$) | *                      | *                       | 33285 (± 203.6)          |
>
> Note: The asterisks (*) indicate that experiments were not required for these configurations. The MPC baseline is independent of the training dataset size. For our Inv-Bi-level approach, it effectively recovers the underlying cost function with minimal data, rendering larger datasets redundant.
>
> 2.  *Can the proposed approach handle deviations from the strong convexity and affine constraint assumptions in practical control tasks?*
>
>     - **Assumptions Clarification:** We would like to clarify our assumptions about the objective function and dynamics. For the objective function, strong convexity is not required. Instead, we include small regularization terms when the objective function is not strongly convex to improve forward stability. For dynamics, our theoretical analysis applies to general control-affine systems rather than being restricted to linear dynamics. This substantially broadens the applicability of our results, as control-affine systems constitute a widely practically important class of dynamical systems [3][4][5].
>     - **Non-convexity:** For some tasks with non-convex feasible region (e.g., obstacle avoidance), the problem can be reformulated as a mixed-integer programming (MIP). There is a growing body of prior work that examines inverse optimization problems in MIP (refer to Section 3.4 in [2]). For the theoretical aspect, our use of ReLU terms to guarantee inverse feasibility extends naturally to MIP settings since the added ReLU terms reshape the objective so that an expert point inside the convex hull can still be made the optimum.
>     - **Empirical Evidence:** Specifically, our Mobile Robot Navigation task inherently involves a non-linear objective function and non-convex feasible region due to collision avoidance constraints. In this context, we leverage the KKT conditions as necessary criteria to identify a solution with strong interpretability. Our experimental results confirm that this framework remains effective in these more general settings.
>
> [1]. Gammelli et al., *Graph Reinforcement Learning for Network Control via Bi-Level Optimization*, arXiv:2305.09129, 2023.
>
> [2]. Chan et al., *Inverse optimization: Theory and applications*, Operations Research, 2025
>
> [3]. Schmitz et al., *On excitation of control-affine systems and its use for data-driven control*, arXiv:2511.03734, 2025
>
> [4]. Kazemian et al., *Random Features Approximation for Control-Affine Systems*, Proceedings of the Conference on Learning for Dynamics and Control, 2024
>
> [5]. Elamvazhuthi et al., *Optimal Transport of Nonlinear Control-Affine Systems*, Proceedings of the IEEE Conference on Decision and Control, 2019

---

> > ### Author Response · Authors · 2025-11-26
> > **Response to Reviewer 4BTY [2/2]**
> >
> > 3.  *What methodology and experimental setup are used for the upper-level reinforcement learning in the proposed hierarchical framework?*
> >
> >     To ensure reproducibility and clarity, we have provided the detailed experimental setup for the upper-level RL policies across all three environments:
> >
> >     - **Autonomous Vehicle Rebalancing**: For the Autonomous Vehicle Rebalancing experiment, we train an A2C-GNN agent as in [1] using on-policy updates with a discount factor of $0.99$, a maximum of $50$ decision steps per episode, and $10000$ training episodes. The actor and critic share the same EdgeConv-based GNN encoder and each head is implemented as a two-layer multilayer perceptron with $256$ hidden units, and both networks are optimized with the Adam optimizer with a learning rate of $5\times 10^{-5}$. At the end of each episode we normalize the discounted returns, perform a single A2C update of actor and critic parameters with gradient-norm clipping at $5$ to improve stability, and we save the checkpoint achieving the highest cumulative training reward for later evaluation.
> >
> >     - **Inventory Management**: We adopt a graph neural network-based Advantage Actor--Critic (A2C) method as in [1] to learn the control policy. Training is performed with an on-policy A2C update, using a discount factor of 0.99, a maximum episode length of 30 steps, and up to 20,000 training episodes. Both actor and critic are optimized with Adam with learning rate 5e-5, and gradient norm clipping is applied to improve training stability.
> >
> >     - **Mobile Robot Navigation**: For the mobile robot, we use an off-policy Soft Actor--Critic (SAC) algorithm. SAC is trained with a replay buffer of size $10^6$, a mini-batch size of $256$, Adam optimizer with learning rate $3\times 10^{-4}$ for both actor and critic, discount factor $\gamma = 0.99$, soft target network updates with $\tau = 0.005$, and automatic entropy tuning with target entropy set to $-\text{dim}(\mathcal{A})$. Gradient updates start once the replay buffer contains more than $400$ transitions, and the agent is trained for $500$ episodes, periodically saving the actor network during training.
> >
> >    We have added the above mentioned RL methodologies and experimental settings to the appendix.
> >
> > [1]. Gammelli et al., *Graph Reinforcement Learning for Network Control via Bi-Level Optimization*, arXiv:2305.09129, 2023.

---

> ### Author Response · Authors · 2025-11-29
> **Acknowledging Score Increase (4->6) Prior to OpenReview Bug**
>
> We sincerely thank you for upgrading your rating **from 4 to 6** in response to our rebuttal, **prior to the recent OpenReview bug**. We noted that this score change was made **without a written reply**. We appreciate your recognition of our paper and response.

---

### Author Response · Authors · 2025-12-03
**General Comment - Rebuttal Summary**

We thank the reviewers for their constructive feedback. The primary focus on our reponses can be summarized as follows:

**1. Clarifying scope and generality of the approach** [Reviewer 4BTY, 2; Reviewer Vxfa, 6 ]

We clarified that our analysis holds for general control-affine dynamics with linear or quadratic objectives, which already covers a broad and practically important class of systems. Moreover, we highlight the potential for extending our framework to more complex settings, both theoretically and empirically, for the following reasons:

- **From a theoretical perspective:**

  (a) *Inverse feasibility:* For a more general class of problems (e.g., tasks with non-convex feasible regions or combinatorial complexity), the problem can be reformulated as a mixed-integer program. There is a growing literature on inverse optimization for MIPs. Our use of ReLU terms to guarantee inverse feasibility extends naturally to such MIP settings, since the ReLU-augmented objective can still make an expert point in the convex hull optimal.

  (b) *Forward stability:* Regarding forward stability argument, Corollary 1 extends to general strongly convex functions, not just quadratic ones.

- **From an empirical perspective:**

  (a) *Support from modern IO literature:* While inverse feasibility cannot always be guaranteed in fully general nonlinear problems, modern data-driven inverse optimization formulations that minimize optimality gap provide statistical consistency guarantees, offering a principled path to extending our framework.

  (b) *Empirical evidence:* Our mobile robot navigation task has a nonlinear objective and non-convex feasible region due to collision avoidance. In this setting, we use a data-driven optimization method and KKT conditions as necessary criteria to obtain interpretable solutions, and empirically observe that our framework continues to perform well in this more general regime.

**2. Additional experiments and baselines**

- **Imitation learning:** [Reviewer 4BTY, 1] We added imitation-learning baselines trained on varying numbers of expert trajectories; our bi-level method consistently outperforms them and remains strong even with few demonstrations indicating good sample efficiency.
- **Inverse MPC baseline:** [Reviewer UqDh, 6] We further implemented a modern Inverse MPC baseline [1] (and a variant where its learned cost is embedded into our bi-level architecture); in both cases, our method attains higher reward, underscoring the benefit of the proposed ReLU-based cost.
- **Sensitivity of RL algorithm:** [Reviewer UqDh, 5] For the mobile robot task, a reward-shaping sensitivity study shows that performance is stable and method ranking is unchanged, suggesting that our conclusions are robust.

**3. Reporting Scalability and sensitivity**

- **Scalability.** [Reviewer UqDh, 4; Reviewer Vxfa, 2&3] We highlighted scaling curves for the inverse-optimization stage, varying problem dimension, number of ReLU terms, and number of demonstrations. The results show predictable growth in solve time. Importantly, all IO computation is offline and does not impact real-time control.
- **Sensitivity.** [Reviewer btwC, 2] We extended our theoretical analysis (Proposition 4) to quantify how noise in expert actions affects the learned lower-level cost. And we report sensitivity to the number of ReLU terms in the Appendix C.4.


**4. Writing improvements to updated manuscripts**
Based on reviewers' feedback, we have incorporated the following changes into our updated manuscripts:

- We enriched the introduction and related work with additional citations and clearer positioning. [Reviewer t9Sj, 1; Reviewer UqDh, 1]
- We added specified upper-level RL details (algorithms, architectures, hyperparameters, and evaluation protocol) in the Appendix C.1. [Reviewer 4BTY, 3; Reviewer UqDh, 5]
- We reported the solve-time scaling results in the Appendix C.5.
- We expanded the discussion on the motivations for inverse optimization from both theoretical and empirical perspectives in the Appendix C.2. [Reviewer t9Sj, 6]

Finally, We would like to briefly mention that the first reviewer 4BTY had already increased their score **from 4 to 6** prior to the **OpenReview bug**, but did not post a reply; thus, this update is now difficult to verify in the system, except through my written response to that reviewer.

In addition, the other four reviewers have remained silent, likely due to time limitations (OpenReview bug occurred only **one day** after I submitted my responses). Regrettably, the current mechanism now precludes them from responding or adjusting their scores after I resolved some **key concerns** (e.g., the misunderstandings regarding baseline naming [Reviewer ts9j, 8; Reviewer UqDh, 2]). Thank you for your time and for considering these details.

[1]. Zhang et al., *Inverse Model Predictive Control: Learning Optimal Control Cost Functions for MPC*, IEEE Transactions on Industrial Informatics, 2024.

---

### Meta-Review · Area_Chair_LK5e · 2026-01-09

**Summary:**

The paper proposes a hierarchical control framework where the lower-level policy is driven by rewards obtained via inverse optimal control from a small number of demonstrations for the high-level task, ensuring that the lower-level objective is aligned with the high-level goal.

In my opinion, the reviewer's biggest initial concerns were:

- The approach isn't, strictly speaking, RL, since it relies on demonstrations to induce the lower-level reward, and needs to be compared to imitation learning baselines.

- The approach makes strong structural assumptions about lower-level problems: convexity and affine constraints.

- The higher-level RL approach used in the experiments isn't sufficiently described in the paper.

- (Alleged) lack of comparison to hierarchical methods.

- Experiments are only on simulated, artificial environments.

- The approach seems limited to low-dimensional action spaces (100D or less)

The rebuttals, in my opinion, have addressed most of these well, other than the issue of structural assumptions about the low-level problem and the experiments being limited to synthetic problems.

I believe that conceptually, the paper offers an elegant, well-founded formulation of hierarchical RL, even if RL isn't strictly the right term here due to the need for expert demonstrations. Unfortunately, though, the two aforementioned issues that remain unaddressed in the rebuttals (not for the lack of the authors' efforts) are inherent to the proposed approach and significantly limit its applicability:

- The reliance on (structured) inverse optimal control methods prevents applying the paper's approach to problems where the inputs are high-dimensional and unstructured, e.g., images. For instance, it's entirely unclear how to use it for robotic manipulation, one area where an approach like this could be very useful but where the main observations come from video cameras.

- This goes beyond robotic manipulation: pretty much the entire agentic decision-making, which relies on text as the main modality, is beyond the method's scope. So, it's non-obvious what major non-synthetic problems this method could actually solve *and* where its ability for high-level planning would be needed. Indeed, the paper doesn't appear to provide any examples of real problems this method is meant to solve.

Overall, the paper's technical contribution is interesting and has its audience, but it's not a very good fit for a conference like ICLR, which focuses on learned and hence usually unstructured representations, to which the proposed method hasn't been demonstrated to apply.

**Reviewer Concerns:**

Please see above.

**Reviewer Scores:**

Reviewer 4BTY initially gave a 4 but, according to the authors, later raised their score to 6 without posting a response. I am willing to believe this, as the rebuttal reasonably addressed most of their concerns.

Reviewer UqDh gave a 6. The rebuttal addresses their concerns well, but it's doubtful whether that would make the reviewer sufficiently excited to raise the score to 7.

Reviewer t9Sj gave a 2. A lot of their concerns were meaningfully addressed by the rebuttal, and I could easily see them raising their score to 4, but probably not higher, due to the approach's limitations that I described above and would surface during the discussion with reviewers if it took place.

Reviewers Vxfa and btwC gave a 4 and a 6, respectively. While the rebuttal should have addressed some of their concerns, the narrow scope of applicability and reliance on strong assumptions about problem structure remain outstanding, which would probably prevent these reviewers from raising the scores.

---

### Decision · Program_Chairs · 2026-01-26

Reject